# Prediction of nonlayered oxide monolayers as flexible high-κ dielectrics with negative Poisson's ratios

Yue Hu ⬡[1] ✉, Jingwen Jiang[2], Peng Zhang ⬡[1,3], Zhuang Ma[1], Fuxin Guan[4], Da Li[1], Zhengfang Qian[1,3], Xiuwen Zhang ⬡[5,6] ✉ & Pu Huang ⬡[1,3] ✉

During the last two decades, two-dimensional (2D) materials have been the focus of condensed matter physics and material science due to their promising fundamental properties and (opto-)electronic applications. However, high-κ 2D dielectrics that can be integrated within 2D devices are often missing. Here, we propose nonlayered oxide monolayers with calculated exfoliation energy as low as 0.39 J/m² stemming from the ionic feature of the metal oxide bonds. We predict 51 easily or potentially exfoliable oxide monolayers, including metals and insulators/semiconductors, with intriguing physical properties such as ultra-high κ values, negative Poisson's ratios and large valley spin splitting. Among them, the most promising dielectric, $GeO_2$, exhibits an auxetic effect, a κ value of 99, and forms type-I heterostructures with $MoSe_2$ and $HfSe_2$, with a band offset of ~1 eV. Our study opens the way for designing nonlayered 2D oxides, offering a platform for studying the rich physics in ultra-thin oxides and their potential applications in future information technologies.

Two-dimensional (2D) materials were used as semiconducting channels or metallic electrodes in the 2D field-effect transistors (FETs) that have been the focus of materials science and device physics in the last decades[1–8]. The gate dielectrics of the 2D FETs have been realized by transferring thin films of bulk oxides onto 2D semiconductors[9], deposition of high-κ oxides on 2D materials[10], or oxidation of the 2D semiconductors into their native oxides[11]. On the other hand, 2D high-κ oxides could be easily transferred onto the 2D semiconducting channels, offering optimal compatibility with 2D semiconductors as well as the opportunity to realize the pure-2D FET that is based solely on 2D crystals. Furthermore, the potentially abundant physical properties in the 2D gate dielectrics combined with the mechanical metamaterials[12] or valleytronic systems[13] could significantly enrich the functionalities of the 2D FETs. However, the currently known 2D material space is rather limited[14–16], which constrains the progress of 2D devices.

A vital progress towards searching for 2D materials including oxides was made by Mounet et al. where 5619 layered bulk precursors were screened from 108,423 experimentally known 3D compounds, and 1825 of these were identified to be potentially exfoliable using density-functional theory (DFT) calculations, including only a few oxide semiconductors and oxide metals[17]. Obviously, layered bulk precursors are limited in the bulk material space. Nonlayered bulk precursors, by contrast, are abundant and untapped, so if they could be exfoliable, nonlayered crystals would become a vast and diverse source of 2D materials. However, although many efforts have been made to find the possible exfoliation method for nonlayered materials[18–21], such as liquid metal-assisted exfoliation, cryogenic exfoliation and gel-blowing exfoliation, exfoliating nonlayered crystals has been a significant challenge, hindering the exploitation of 2D materials.

[1]Key Laboratory of Optoelectronic Devices and Systems of Ministry of Education and Guangdong Province, College of Physics and Optoelectronic Engineering, Shenzhen University, 518060 Shenzhen, China. [2]School of Information Engineering, Jiangmen Polytechnic, Jiangmen, China. [3]State Key Laboratory of Radio Frequency Heterogeneous Integration, Shenzhen University, 518060 Shenzhen, China. [4]Department of Physics, University of Hong Kong, Hong Kong, China. [5]College of Physics and Optoelectronic Engineering, Shenzhen University, 518060 Shenzhen, China. [6]Present address: Renewable and Sustainable Energy Institute, University of Colorado, Boulder, CO 80309, USA. ✉e-mail: yuehuphd@szu.edu.cn; xiuwenzhang@szu.edu.cn; arvin_huang@szu.edu.cn

To address this challenge, we propose a route to design 2D oxides from nonlayered precursors. This involves geometric screening of promising exfoliated crystallographic planes from nonlayered oxides and conducting van der Waals (vdW) DFT calculations of the exfoliation process. Our results demonstrate that under the external exfoliation stress, the close-packed planes of the continuously bonded oxides can be peeled off from the nonlayered precursors, and reconstructed to form stable monolayer structures. Such an exfoliation process of strongly bonded oxides could have an exfoliation energy as low as 0.39 J/m$^2$, comparable with that of the common 2D vdW materials (e.g., graphite[17,22]), which stems from the ionic feature of the metal oxide bonds and the flexible coordination numbers of the metal atoms by the ligands during the reconstruction process. We predict 51 easily/potentially exfoliable stable oxide monolayers, including 10 metals and 41 insulators/semiconductors (band gaps, 0.5 to 6.8 eV), with interesting physical properties such as high $\kappa$ value (99), negative Poisson's ratios (NPRs, −0.094 to −0.431), as well as large valley spin splitting (211 meV). Among the screened 2D oxide dielectrics, GeO$_2$ monolayer exhibits a wide band gap of ~3.3 eV and an unusually high $\kappa$ value (99), which is much higher than that of the bulk GeO$_2$ (~6)[23], as well as the 2D dielectrics CaF$_2$ (~6)[24] and β-Bi$_2$SeO$_5$ (~22)[25]. We further explore the physical properties of the hetero-bilayers (HBLs) that consist of the predicted best candidate 2D oxide dielectric GeO$_2$ monolayer and 2D transition-metal dichalcogenides. Type-I GeO$_2$/MoSe$_2$ and GeO$_2$/HfSe$_2$ 2D vdW HBLs with band offset of ~1 eV are validated, which are highly promising for 2D FET applications. Our study proposes a route to exfoliate highly desired 2D functional materials from nonlayered bulk crystals, significantly expanding the material space of available dielectrics for flexible 2D electronic and optoelectronic applications.

## Results

### Nonlayered oxides exfoliation

We start with a set of nonlayered binary metal oxides in Materials Project[26] (see Supplementary Information section Workflow and Supplementary Fig. 1 for filtering details) and then utilize geometric criteria to identify the promising exfoliated crystallographic planes. The criteria are based on the packing ratio of in-plane and out-of-plane, which identify the crystallographic planes with large interplanar spacing and in-plane close packing. These planes exhibit a relatively weak interplanar interaction, along with a significant anisotropy between the interplanar and in-plane bonding, enabling exfoliation from the bulk rather than fracture. The close-packed plane is described in detail in the Methods section. The geometric criteria provide a cost-effective way to quickly identify the promising exfoliated planes from a large number of nonlayered oxides, significantly reducing the amount of DFT calculations. In addition, the criteria are independent of chemical composition, bonding type, and symmetry, and hence can be extended to any nonlayered crystal. Guided by the criteria, we identify 97 promising exfoliated crystallographic planes from 47 nonlayered oxides, including close-packed and nearly close-packed planes (a close-packed plane often has a large interplanar spacing, see Methods). The promising exfoliable planes identified based on the geometric criteria are further evaluated through vdW-DFT calculations of their interplanar binding energies ($E_b$).

The procedure for calculating $E_b$ is illustrated schematically in Fig. 1a–d. Please refer to the Supplementary Information section Nonlayered Oxides Exfoliation as well as Supplementary Fig. 2 for a detailed explanation of the exfoliation procedure. The exfoliation processes for all the considered materials can be summarized as follows: as the external exfoliation stress gradually increases, the interlayer bonding interactions gradually weaken until separation, followed by the surface reconstruction of each separated part, eventually forming a 2D structure counterpart that does not change with stretching, as shown in Fig. 1b, c. The corresponding $E_b$ curves are shown in Fig. 1a, d. We note that in the first stage of exfoliation, namely

from 3D precursor to separation, the system energy increases gradually for all materials under consideration. However, during the surface reconstruction phase, there are two types of energy evolution: if there is significant structural deformation in the surface reconstruction process, such as the formation of intralayer chemical bonds, as seen in Fig. 1c, the system energy usually decreases significantly, as shown in Fig. 1d; while if the 2D structure after surface reconstruction still roughly can be found the counterpart from its 3D precursor, as seen in Fig. 1b, then the system energy in the surface reconstruction phase usually continues to increase and eventually converges, as shown in Fig. 1a, which is similar to that observed in layered materials[22].

We compare the $E_b$ of close-packed planes with that of nearly close-packed planes (Supplementary Fig. 4). As expected, most of the close-packed planes exhibit lower $E_b$ than the nearly close-packed planes, validating the geometric criteria. Moreover, in cases where the $E_b$ of nearly close-packed planes is lower, they usually have wider interplanar spacings than the close-packed planes. To summarize, a crystal plane exhibits low $E_b$ when it satisfies both in-plane close packing and wide interplanar spacing. In contrast, if the crystal plane satisfies one of the two conditions, additional DFT calculations are necessary to determine the feasibility of its exfoliation. These results demonstrate that the geometric criteria can accurately identify promising exfoliated crystallographic planes among a large number of crystals at a low cost. This allows for DFT calculations of nonlayered oxides exfoliation, and provides a foundation for further investigation into the rich physics of 2D nonlayered oxides.

We plot in Fig. 1e $E_b$ versus interplanar spacing $d_{hkl}$ for 97 2D oxides exfoliated from 47 nonlayered bulks. The corresponding values are listed in Supplementary Table 3. Notably, the inverse relation between $E_b$ and $d_{hlk}$ is clearly noticeable. Most materials with low $E_b$ exhibit large $d_{hlk}$, distributing in the bottom right corner in Fig. 1e; while most materials with high $E_b$ exhibit small $d_{hlk}$, concentrating in the top left corner in Fig. 1e. This is expected, since large interplanar spacing usually indicates weak interplanar bonding. An extreme example, layered materials have relatively large interplanar spacings, whose interlayer interactions are weak vdW forces accordingly.

To further identify the exfoliable oxides, we choose thresholds of 1.0 J/m$^2$ for $E_b$, and classify candidates falling below this threshold as "easily exfoliable" (EE), marked by red in Fig. 1e. This choice has been used to identify the layered 2D materials commonly exfoliated in experiments[27]. The results show that although most binding energies are higher than this threshold value for the considered oxides, there are still 17 candidates belonging to the EE region. Most of these oxides are metallic, with the possible applications as electrodes and naturally avoid the oxidation problem. Furthermore, a compelling 2D oxide, WO$_3$ (110), is found in this region, exhibiting anomalous mechanical property. Specifically, the monolayer expands laterally regardless of whether longitudinal stretch or compression is applied. In the top left corner of Fig. 1e, a number of materials denoted with green circles present high $E_b$ (up to 10.3 J/m$^2$) and small $d_{hlk}$, which forms the upper limit of 3 J/m$^2$ for the exfoliation process. Above this value, materials are difficult to exfoliate and are thus excluded from further investigation of physical properties. The remaining materials (blue circle) possess relatively weak bonding. We classify oxides belonging to this group as "potentially exfoliable" (PE), where the HfO$_2$ ($\bar{1}$11) monolayer has been synthesized in experiment[19]. Additionally, several intriguing 2D functional oxides are discovered in this region, including 2D high-$\kappa$ oxide GeO$_2$ with NPR, 2D antiferromagnetic (AFM) high-$\kappa$ oxide Fe$_2$O$_3$, 2D valleytronic oxide BaO (111), and various auxetic oxide monolayers.

To assess the structural stability of both EE and PE 2D oxides, we perform phonon dispersion calculations. Initially, we categorize the 2D oxides based on their space groups (Supplementary Tables 5–20) and select representative structures from each group for the calculations. Before conducting specific calculations, we determine the magnetic ground states of the 2D oxides that originate from the magnetic bulk

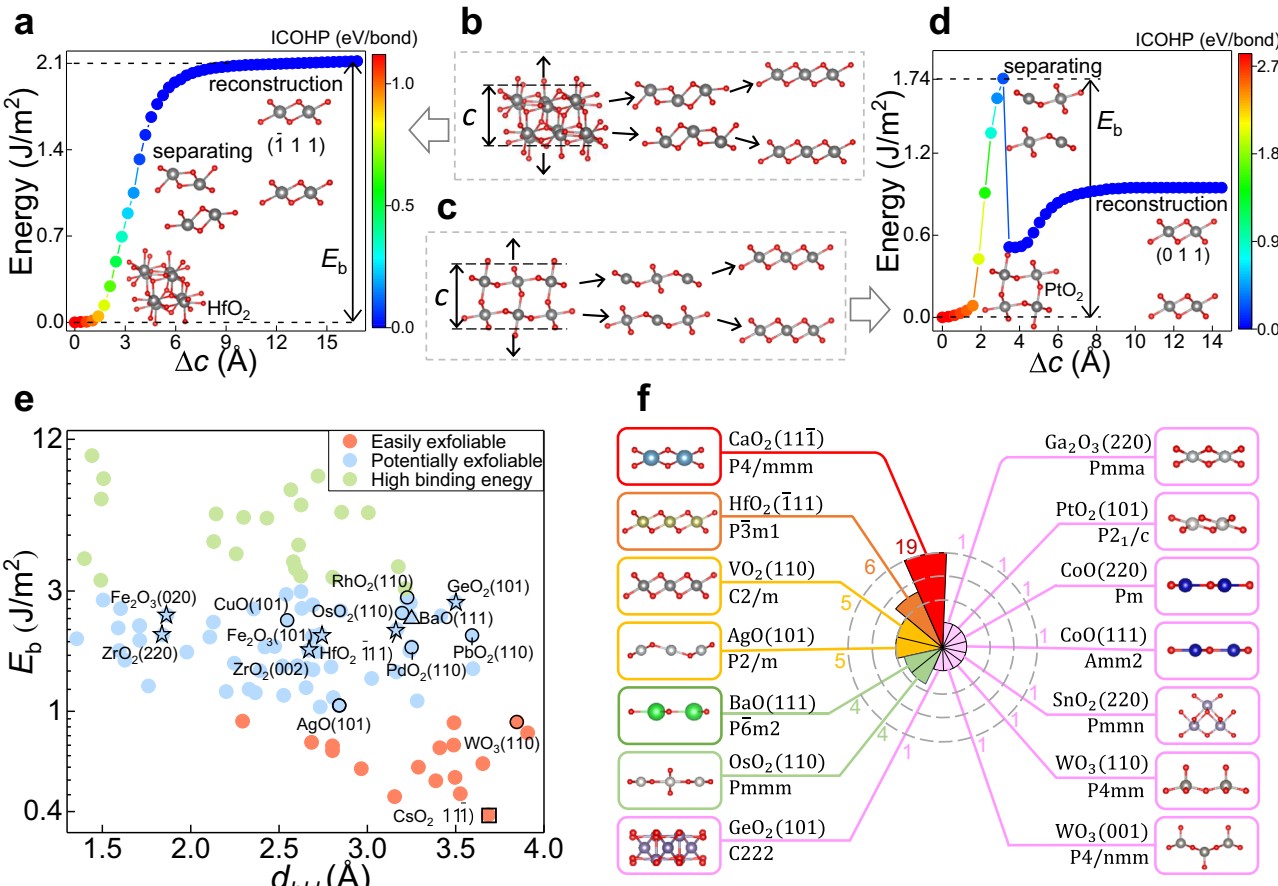

**Fig. 1 | Schematic of exfoliation procedure and summary of exfoliation results for nonlayered binary metal oxides. a, d** Interplanar binding energy ($E_b$) versus the change in the lattice constant $c$ ($\Delta c$). Each point is color-coded based on the absolute value of the integrated crystal orbital Hamilton population (ICOHP). The higher the ICOHP value, the stronger the bonding strength. Insets show the atomic structures of $HfO_2$ and $PtO_2$ under external stretching, from bulk to separation, to reconstruction to form a stable 2D structure. **b, c** The procedure for calculating $E_b$. **e** $E_b$ versus interplanar spacing ($d_{hkl}$). Materials classified as easily exfoliable, potentially exfoliable, and high binding energy are demonstrated in different colors. The selected 2D oxide dielectrics (five-pointed star), auxetic monolayers (circle), 2D valleytronic oxide (triangle), and the monolayer with the lowest $E_b$ (rectangle) are marked by different symbols and chemical formulas. **f** Polar histogram for easily/potentially exfoliable 2D stable oxides classified according to space groups. The crystal structure and chemical formula of representative oxide for each space group is given, as well as the total number of 2D oxides contained in the space group. Red spheres denote O atoms and the other colored spheres indicate metal atoms.

precursors by comparing the total system energy of ferromagnetic (FM) and all possible AFM orderings of on-site spins in a cell containing at least four transition-metal atoms. This process enables us to identify 8 AFM and 2 FM 2D oxides (Supplementary Table 4). The computed phonon dispersions, shown in Supplementary Figs. 5–30, reveal that most of the oxide monolayers presented non-imaginary frequency phonon dispersions, indicating dynamic stability, while the remaining about 21% of the monolayers exhibit negative phonon spectra, predicted to be unstable. For these unstable monolayers, one could search for their stable or metastable structures by investigating the phase-transition pathway[28]. However, due to workload constraints, we exclude the unstable monolayers from further study in this work.

Subsequently, we thoroughly evaluate the actual stability of 2D oxides with stable phonon dispersion. We initiate this process by performing ab initio molecular dynamics (AIMD) calculations. The results shown in Supplementary Figs. 6–8, 10–16, 20–24, 26–30 show that for most of the monolayers, the total energy fluctuation throughout the simulation is rather small, and the final structures do not shatter and are only distorted a little, demonstrating that they are thermally stable. However, there are exceptions such as the $Ag_2O$ (111) prototype and $VO_2$ (022), whose final structures are evidently distorted and cannot be relaxed back to their initial structures, which are

not included in the list of predicted stable 2D oxides. Additionally, we calculate the hydrogen binding energy ($E_{bind}$ [$H_2$]) on the monolayers to assess their surface inertness. The results presented in Supplementary Fig. 31 indicate that the $E_{bind}$ [$H_2$] of our predicted 2D oxides, especially our highly promising 2D functional oxides such as $GeO_2$, $Fe_2O_3$, $WO_3$, $RhO_2$, $OsO_2$, etc., ranges from −0.038 to −0.135 eV/$H_2$, which is moderate and comparable to that of common 2D TMDs (−0.063 eV/$H_2$). Monolayers with $E_{bind}$ [$H_2$] outside this range, namely $SrO_2$, $CaO_2$, $VO_2$, and $MnO$, have been removed from the list of predicted stable 2D oxides. Finally, for our most promising 2D high-$\kappa$ dielectric oxides, namely $GeO_2$, $Fe_2O_3$, $WO_3$, $RhO_2$, and $SnO_2$, we perform AIMD simulations at room temperature under various environments containing water, oxygen, or hydrogen. The results, shown in Supplementary Fig. 32, further confirm their ability to exist stably under ambient conditions.

We plot the polar histogram in Fig. 1f to show the predicted stable EE and PE 2D oxides, comprising 14 space groups that include non-centrosymmetric and polar as well as chiral structures (Supplementary Table 22). The most common space group is P4/mmm, containing $CaO_2$ (11$\bar{1}$) and 18 similar structures. Some space groups like P$\bar{3}$m1, P$\bar{6}$m2 and Pmmn can also be commonly found in layered 2D materials[17,29]. Besides, the graphene-like monolayer[30–33], such as ZnO

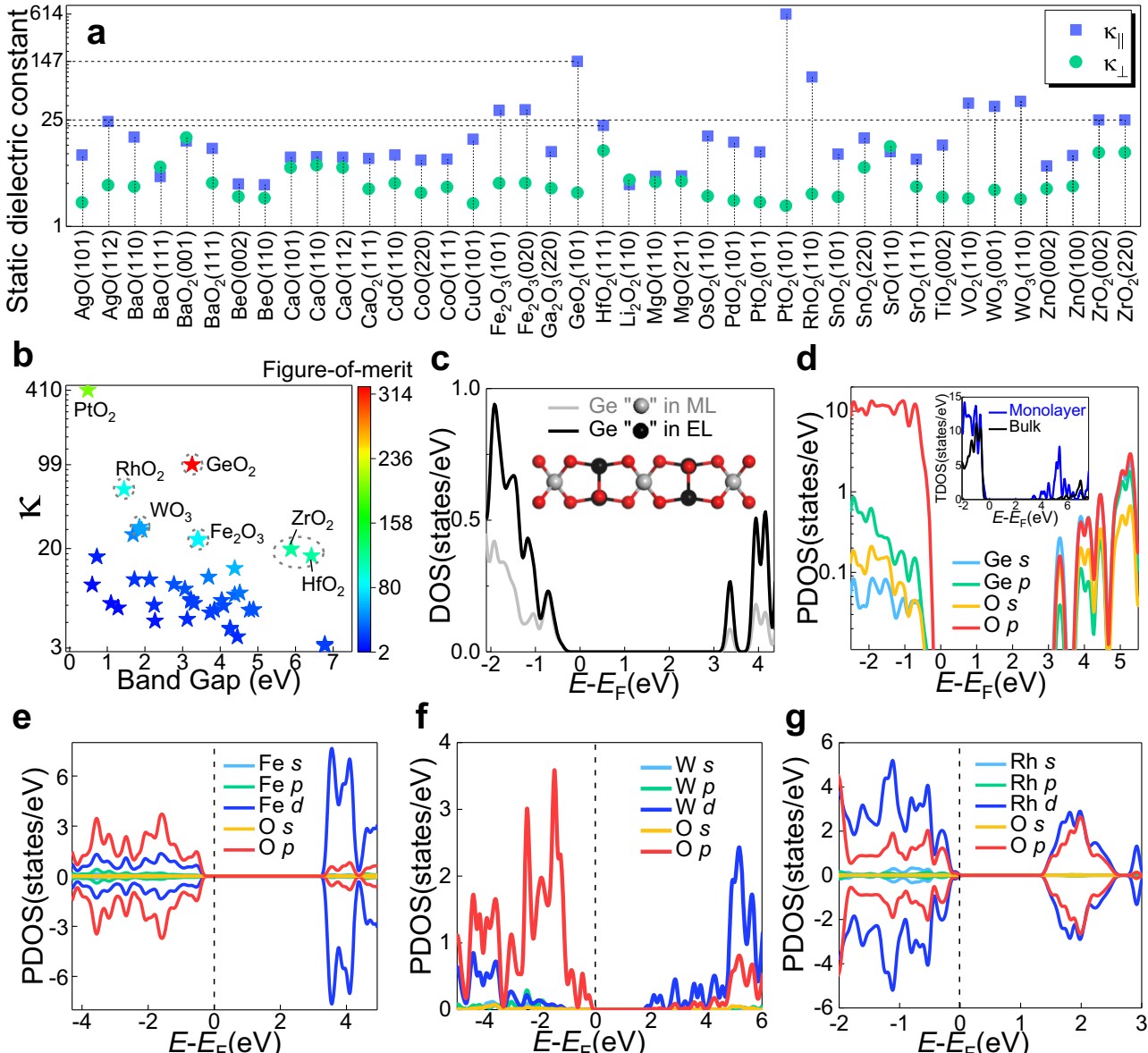

**Fig. 2 | Dielectric properties for 2D oxide insulators/semiconductors. a** In-plane (∥) and out-of-plane (⊥) static dielectric constants, $\kappa$. **b** $\kappa$ versus band gap ($E_g$). Each point is color-coded according to the figure-of-merit, $E_g \cdot \kappa$. $GeO_2$ (101) appears as the most promising predicted material. **c** Comparison diagram for total density of states (TDOS) of Ge in the middle layer (ML) and Ge in the edge layer (EL). The inset shows the crystal structure of $GeO_2$ (101). Projected density of states (PDOS) for **d** $GeO_2$ (101), **e** $Fe_2O_3$ (101), **f** $WO_3$ (110), and **g** $RhO_2$ (110) monolayers show strong cross-gap hybridization between the O $p$ valence bands and the corresponding metal conduction bands. Inset in **d**: Comparison for TDOS of 2D and bulk $GeO_2$.

(002), BeO (002)/(110), and BaO (111), have already been suggested as exfoliable, supporting our exfoliation procedure.

Encouraged by the aforementioned outcomes, we conduct a one-by-one comparison between our stable 2D oxides and the structures in the C2DB[34] database. This involves comparing the chemical formula, lattice parameters, space group, atomic position, and total system energy. The detailed comparison procedure is given in the Methods section. Our predicted stable 2D oxides correspond to 32 chemical formulas, out of which 12 can be found in the C2DB database. The comparison results, presented in Supplementary Table 21, indicate that out of the 12 chemical formulas, 10 have at least one structure with energy equal to or lower than the lowest energy structure in the C2DB database. Notably, there are 8 identical structures, which validates the reliability of our exfoliation procedure. Additionally, we emphasize that the multi-functional 2D oxides, such as auxetic high-$\kappa$ oxides ($GeO_2$, $RhO_2$, $WO_3$), AFM high-$\kappa$ oxide ($Fe_2O_3$), and 2D valleytronic

oxide (BaO) that we discuss below, are structures not present in the C2DB database.

## 2D oxide dielectrics screening

For the aim to expand the family of 2D oxide dielectrics, we calculate the electronic band structures of stable EE and PE 2D oxides with the HSE functional, obtaining 41 2D oxide insulators/semiconductors with band gaps ranging from 0.5 to 6.8 eV (Supplementary Tables 22 and 23), and then further calculate their static dielectric constants. Both the ionic and electronic contributions to the dielectric response are computed. Figure 2a displays the in-plane and out-of-plane static dielectric constants of 2D oxide insulators/semiconductors. The corresponding values are listed in Supplementary Table 23. Obviously, $PtO_2$ (101) and $GeO_2$ (101) monolayers exhibit the highest in-plane dielectric constants (614, 147), which are higher than that of 2D dielectric TlF (98.4)[29], the maximum value as far as we are aware. In the

out-of-plane direction, the highest dielectric constant (15) belongs to $BaO_2$ (001) monolayer. We plot the comparison charts between the ionic and electronic contribution in the in-plane and out-of-plane directions, respectively, in Supplementary Figs. 33 and 34. In the in-plane direction, the dielectric responses for most of 2D oxides are dominated by the ionic contribution, while in the out-of-plane direction, the electronic components are higher. Besides, we also find that the ionic contribution differs greatly in-plane and out-of-plane directions, but not greatly for the electronic contribution, see supplementary Table 23, suggesting that the marked difference between the in-plane and out-of-plane static dielectric constants primarily arises from the ionic contribution. This trend is similarly observed in 2D rare-earth oxyhalides[29].

In Fig. 2b, we plot the property map of the band gap ($E_g$) versus static dielectric constant ($\kappa$). The value of $\kappa$ is obtained by averaging the diagonal components of the dielectric tensor. Firstly, the inverse relation between $E_g$ and $\kappa$ is roughly valid. This inverse relationship should be expected since the electronic susceptibility is inversely proportional to the energy difference of the transition states if first-order perturbation theory is taken into account, and the latter increases on average with the band gap. Secondly, oxides with large values of both $E_g$ and $\kappa$ are scarce. To select the best 2D oxide dielectrics, we define a figure-of-merit ($f_{FOM}$) that takes into account the fact that the leakage current decreases exponentially with the increase of the figure-of-merit, $f_{FOM} \sim (m_{eff} \bullet \Phi_b)^{0.5}\kappa$, where $m_{eff}$ is the tunneling effective mass of electron or hole, and $\Phi_b$ is the injection barrier[35]. In principle, one can compute $m_{eff}$ and $\Phi_b$ accurately. Here, considering of the computational burden and the fact that there exists a positive correlation between $(m_{eff} \bullet \Phi_b)^{0.5}$ and $E_g$, we simplify $f_{FOM}$ as $E_g \bullet \kappa$[36]. Each point in Fig. 2b is color coded according to $f_{FOM}$.

As shown in Fig. 2b, the figure-of-merit of $GeO_2$ is particularly noteworthy among all the 2D oxide dielectrics. This is due to its ultra-high $\kappa$ value of 99, coupled with an $E_g$ value greater than 3 eV, which makes it highly promising for DRAM applications. It is easy to see from Supplementary Table 23 that the unusually large static dielectric constant of $GeO_2$ is dominated by the ionic contribution, which accounts for about 96%. Besides, we calculate the static dielectric constant of bulk $GeO_2$, including the electronic (~3.3) and ion (~3.4) contributions. Compared to the bulk $GeO_2$, the ion dielectric constant of $GeO_2$ monolayer increases by a factor of about 28, while the electronic dielectric constant does not change much (Supplementary Table 23). Apparently, the anomalously enhanced static dielectric response of $GeO_2$ monolayer is mainly caused by the ionic contribution. Moreover, the total density of states of Ge in the middle and edge layers shown in Fig. 2c reveals that Ge electronic states in the valence and conduction bands are mainly from Ge in the edge layer except for the first peak in the valence bands where the contributions from Ge in the middle and edge layers are essentially equal. The band gap narrowing of the $GeO_2$ monolayer is accompanied by an obvious increase in the number of states near or at the valence/conduction bands maxima/minima compared to the bulk $GeO_2$, as shown in the inset of Fig. 2d. The projected density of states (PDOS) of $GeO_2$ monolayer shown in Fig. 2d shows strong cross-gap hybridization between the O $p$ valence band states and Ge $s$ and $p$ conduction band states. The strong cross-gap hybridization leads to strongly enhanced Born effective charges, similar to that of sulfosalts[37]. The enhanced Born effective charges in the $GeO_2$ monolayers are expected to lead to a dominant ion contribution in the static dielectric constant.

In addition to the $GeO_2$ monolayer, we have also identified several noteworthy candidates, including $Fe_2O_3$ (020)/(101), $WO_3$ (110), and $RhO_2$ (110). $Fe_2O_3$, $WO_3$, and $RhO_2$ have higher dielectric constants than that of $HfO_2$, with $\kappa$ values of approximately 24, 30, and 62, respectively. The PDOSs of the $Fe_2O_3$, $WO_3$, and $RhO_2$ monolayers shown in Fig. 2e–g demonstrate that the O $p$ valence bands and corresponding transition metal (Fe, W, Rh) $d$ conduction bands exhibit strong cross-gap hybridization, which leads to the large static dielectric constants. Additionally, we calculate the magnetic anisotropy of 2D $Fe_2O_3$ (101), and the results show that 2D $Fe_2O_3$ (101) exhibits an out-of-plane easy axis and a magnetic anisotropy energy (MAE) of 50 μeV/Fe-atom (Supplementary Table 24). Further, we employ the mean field theory to estimate the Néel temperature of 2D $Fe_2O_3$ (101), using the formula $3k_B T_N = S(S+1)\sum J$[38,39], where $k_B$ represents the Boltzmann constant, $S$ is equal to 5/2 due to $d^5$ for $Fe^{3+}$, and $J$ (3.45 meV) denotes the exchange strength of one Fe with its six nearest neighbors. The $J$ value is obtained from non-collinear DFT total energy calculations based on the Heisenberg model, using $J = \frac{J_\perp + J_\parallel}{2}$, where $J_{\perp/\parallel} = \frac{E_{FM}^{\perp/\parallel} - E_{AFM}^{\perp/\parallel}}{2N_{NN}S^2}$[40]. Here, $E_{FM}^{\perp/\parallel}$ and $E_{AFM}^{\perp/\parallel}$ represent the total energies for FM and AFM order with the magnetic axis oriented in the out-of-plane/in-plane direction. $N_{NN}$ refers to the number of nearest neighbors. To obtain parameters beyond the nearest neighbor, one can consult the method described in reference[41]. Based on our calculations, we can estimate that the Néel temperature of our predicted 2D $Fe_2O_3$ is equal to 700.61 K, indicating a stable antiferromagnetic state at or above room temperature. The high $T_N$ and benign thermal stability of 2D $Fe_2O_3$ makes it a promising candidate for various high-temperature applications, including catalysis and energy storage. Also, the 2D nature of $Fe_2O_3$ offers a large surface area, which could significantly enhance its catalytic activity. Our first-principles calculations also reveal a type-II $Fe_2O_3/MoS_2$ heterostructure, in which the conduction band minimum and valence band maximum are located at $Fe_2O_3$ and $MoS_2$, respectively (Supplementary Fig. 45). This discovery presents opportunities for optoelectronics and photocatalysis[42,43].

## Anomalous mechanical effect in 2D oxides

To assess the flexibility and resistance ability to the distortions in the presence of strain for the 2D high-$\kappa$ oxides obtained above, we calculate their Poisson's ratios and elastic constants. The relevant calculation details are given in the Methods section. Remarkably, the $GeO_2$ monolayer manifests the interesting out-of-plane negative Poisson's ratio (NPR) effect, which stems from the puckered structures (See Supplementary Information section Mechanical Explanation as well as Supplementary Fig. 36 for details). As shown in Fig. 3a, the $GeO_2$ monolayer expands/contracts in the $z$-direction, if a stretch/compression in the $x$-direction is applied, whose NPR value (−0.094) is about three times than that of black phosphorus (−0.027)[12], indicating that $GeO_2$ monolayer possesses more superior toughness[44], higher shear modulus[45] and stronger indentation resistance[46] compared to black phosphorus.

Inspired by the $GeO_2$ monolayers, we further calculate the Poisson's ratio for the other puckered structures in the EE and PE list. Our calculations show that the $PdO_2$, $PbO_2$, $RhO_2$, and $OsO_2$ monolayers display a more obvious auxetic effect in the out-of-plane direction, see Fig. 3b. Their NPR values are approximately 6-16 times than that of black phosphorus[12]. The deformation mechanism is elaborated in the Supplementary Information section Mechanical Explanation as well as Supplementary Fig. 37. Moreover, we find that $WO_3$ (110), AgO (101), and CuO (101) monolayers expand in the $z$-direction, regardless of whether they are subjected to stretching or compression in the $x/y$-direction, as depicted in Fig. 3c, d. That is, positive Poisson's ratio (PPR) appears during compression and NPR appears during stretching. Notably, at near zero strain, their Poisson's ratios become zero, which is analogous to the zero Poisson's ratio reported in refs. 47,48. Additionally, in contrast to the previous findings on Pd-decorated borophene[49], this mechanical effect observed in the $WO_3$, AgO, and CuO monolayers is intrinsic and arises from their unique puckered structures, which we term the biased-NPR effect. The relevant deformation mechanism is illustrated in the Supplementary Information section Mechanical Explanation as well as Supplementary Figs. 38 and 39.

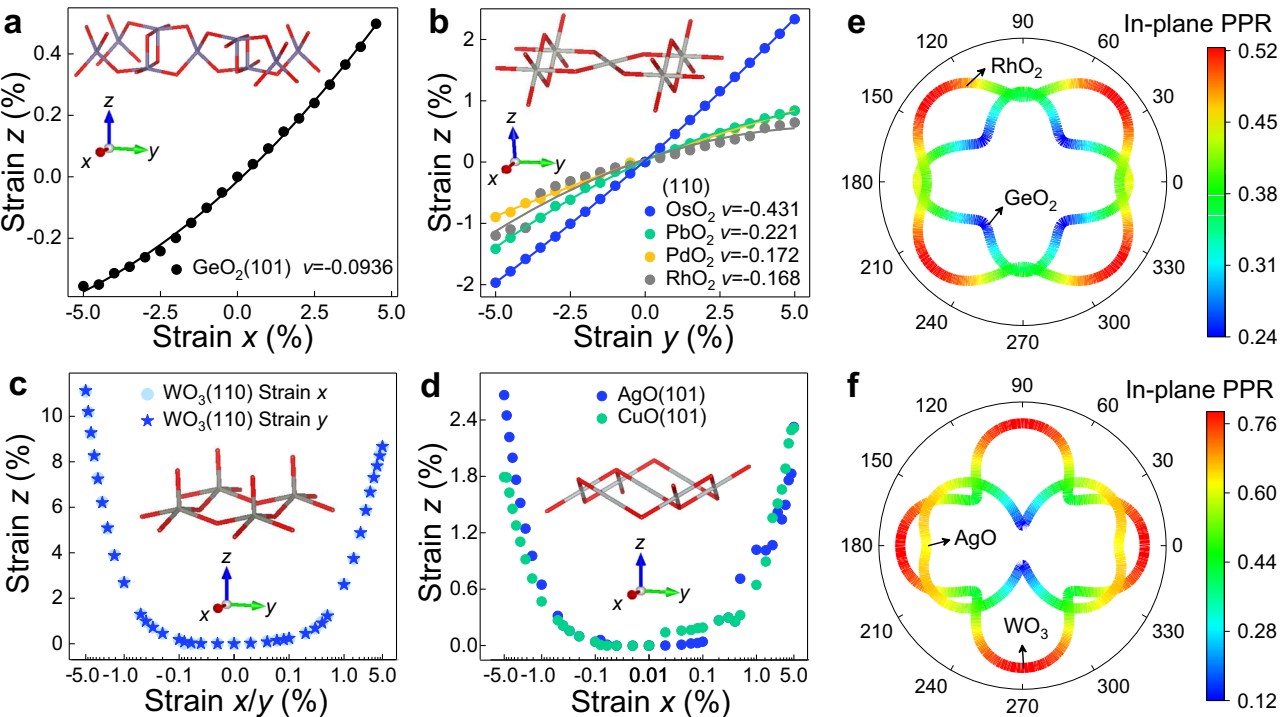

**Fig. 3 | Mechanical properties of 2D auxetic oxides. a** GeO$_2$ (101), **b** MO$_2$ (110) (M = Os, Pb, Pd, Rh) monolayers expand/contract in the $z$-direction, when a stretch/compression in the $x(y)$ direction is applied, exhibiting linear negative Poisson's ratio effect. In contrast, **c** WO$_3$ (110), **d** AgO (101) and CuO (101) monolayers expand in the $z$-direction, regardless if a stretch or compression in the $x(y)$ direction is applied. The insets show the optimized crystal structures, where the red and gray sticks denote oxygen and the corresponding metal atoms, respectively. **e, f** In-plane positive Poisson's ratios (PPR) of GeO$_2$ (101), RhO$_2$ (110), WO$_3$ (110), and AgO (101) as a function of the in-plane angle. 0 degree corresponds to the $x$-axis.

There are two primary deformation mechanisms for 2D materials that exhibit auxetic effects. One is the result of pure geometric effects, such as the out-of-plane negative Poisson's ratio observed in black phosphorus[12]. The other is attributed to the interplay of both geometric and electronic structure effects, observed in transition-metal selenides, transition-metal halides[50], and 1T-type transition-metal dichalcogenides[51]. We note that the auxetic behaviors in our predicted oxide monolayers are independent of chemical elements. Therefore, we attribute the occurrence of out-of-plane NPR and biased NPR effects to purely geometric factors. For further explanations, please refer to the Supplementary Information section Mechanical Explanation.

Further, we take GeO$_2$ (101), RhO$_2$ (110), WO$_3$ (110), and AgO (101) monolayers as examples to calculate their in-plane Poisson's ratios at arbitrary angles. The in-plane Poisson's ratios shown in Fig. 3e, f reveal that the four puckered oxide monolayers possess high anisotropy and positive Poisson's ratios in the in-plane direction. Last, the elastic constants of all the 2D high-$\kappa$ and auxetic oxides satisfy the Born-Huang criteria[52], validating their mechanical stability. The elastic constants and the elastic stability conditions are summarized in Supplementary Tables 27–29.

## 2D valleytronic oxide

We know that spin–orbit coupling can induce spin splitting and spin polarization if the material has sufficiently low crystal symmetry, even in nonmagnetic materials[53]. Here, we characterize the band structures with spin–orbit coupling for all non-centrosymmetric 2D oxides (Supplementary Table 22), and screen out the oxide monolayer BaO (111) that has spin splitting. The electronic band structure of the BaO (111) monolayer in Supplementary Fig. 40(a) shows a clear spin splitting at the K and K′ points in the valence band maximum, which is mainly attributed to the O $p$ orbital, see the PDOS in Supplementary Fig. 41. The splitting reaches around 211 meV, which is much larger

than the splitting size ~148 meV of MoS$_2$[54]. The corresponding spin textures of BaO (111) obtained from all atoms in the unit cell are shown in Supplementary Fig. 40(b, c) and exhibit strong spin polarization. The spin vectors at the energy valley are perpendicular to the $xy$ plane, which is similar to Zeeman-type splitting occurring in WSe$_2$ induced by the electric field[55].

## 2D oxide dielectrics/semiconductors hetero-bilayers

To evaluate the potential of GeO$_2$ in 2D FETs, we investigate the electronic band structures of hetero-bilayers (HBLs) composed of GeO$_2$ and transition-metal dichalcogenides (TMDs) including MoS$_2$, MoSe$_2$, HfS$_2$, and HfSe$_2$. The relaxed lattice constants of GeO$_2$ and TMDs are provided in Supplementary Table 30. We use a 4×2 rectangular supercell of MoS$_2$ (MoSe$_2$) to match a 3×1 GeO$_2$ supercell, and a 2 × 3 rectangular supercell of HfS$_2$ (HfSe$_2$) to match a 3×1 GeO$_2$ supercell. The strains applied to the GeO$_2$ along the $x$-direction for GeO$_2$/MoS$_2$, GeO$_2$/MoSe$_2$, GeO$_2$/HfS$_2$, and GeO$_2$/HfSe$_2$ HBLs are 0.09%, 1.47%, 0.18%, and 0.23%, respectively, and the corresponding values along the $y$-direction are 2.37%, 3.87%, 2.12%, and 2.62%, respectively.

Figure 4a, b shows the optimized structures of HBLs. The calculated interface distance $d_0$ exhibits a similar value of around 0.2 nm (Supplementary Table 31), comparable to that of the artificially assembled vdW interfaces, e.g., Au/MoS$_2$[56], BN/graphene[57], and WSe$_2$/Bi$_2$Se$_3$[58]. The projected electronic band structures of 2D GeO$_2$ (101)/TMD HBLs shown in Fig. 4c, e, g, i can be considered as a rough summation of the band structures of the isolated GeO$_2$ (101) and the corresponding TMD monolayer, indicating a weak van der Waals interaction between them. The band structures and the corresponding band gaps of pristine TMDs are given in Supplementary Fig. 42 and Table 32, respectively, which are in good agreement with literature.

Figure 4d, f, h, j display the band alignments of the GeO$_2$ and TMD monolayers with respect to the vacuum level before and after forming HBLs. It is evident that GeO$_2$ forms a straddling gap or type-I band

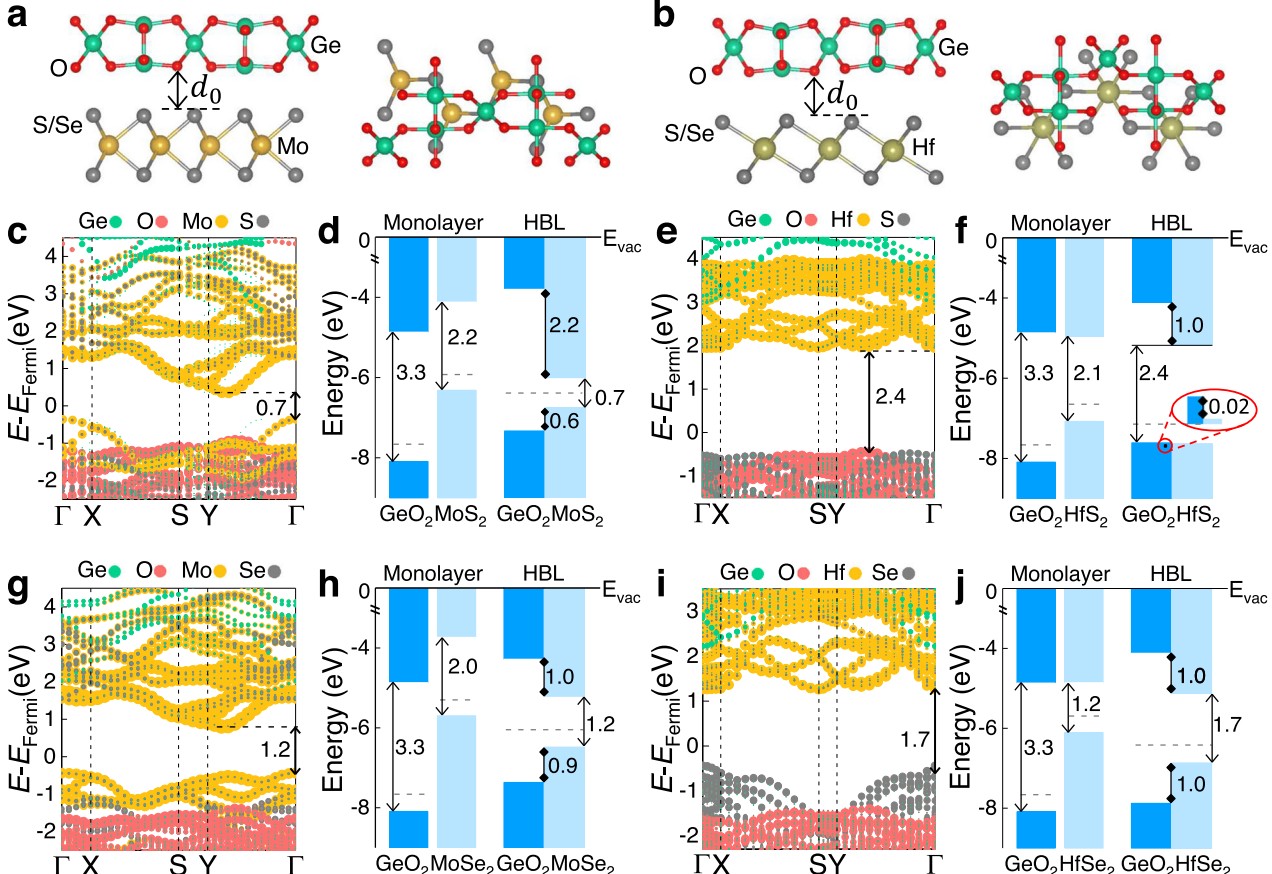

**Fig. 4 | 2D GeO₂/TMDs hetero-bilayers (HBLs).** Side and top views of **a** GeO₂/MoX₂ (X = S, Se) and **b** GeO₂/HfX₂ (X = S, Se), where the interfacial distance $d_0$ between GeO₂ and TMD is marked. Projected band structures of **c** GeO₂/MoS₂, **e** GeO₂/HfS₂, **g** GeO₂/MoSe₂ and **i** GeO₂ /HfSe₂ HBLs, where the high symmetry K-points are indicated by the vertical dashed lines. **d, f, h, j** The band alignment of the GeO₂ and corresponding TMD monolayer before and after forming HBLs, where Fermi level is marked by the horizontal dashed line, and $E_{vac}$ denotes vacuum level. It is easily seen that GeO₂ forms type-I HBLs with MoS₂, MoSe₂ and HfSe₂. Among them, type-I GeO₂/MoSe₂ and GeO₂/HfSe₂ HBLs with band offset of ~1 eV for both CBs and VBs are highly promising for FET applications.

alignment with MoS₂, MoSe₂, and HfSe₂ monolayers. However, in the case of the GeO₂/HfS₂ HBL, the valence band maximum of GeO₂ is only slightly larger than that of HfS₂, by approximately 0.02 eV, which falls within the numerical error range. Therefore, it is not possible to determine whether it is a type-I or type-II alignment. Remarkably, type-I GeO₂/MoSe₂ and GeO₂/HfSe₂ HBLs with about 1 eV band offset between oxide dielectric GeO₂ and channel materials (MoSe₂ and HfSe₂), see Fig. 4h, j, which is basically sufficient to minimize leakage current caused by Schottky emission of carriers into the dielectric, holding great promise for FETs. The other interesting phenomenon in the band alignments is that, compared to the band gaps of the pristine TMD monolayers, the band gaps of GeO₂/MoS₂ and GeO₂/MoSe₂ HBLs decrease, see Fig. 4d, h, while the band gaps of GeO₂/HfS₂ and GeO₂/HfSe₂ HBLs increase, see Fig. 4f, j. The charge arrangement from TMD to GeO₂ is "-+-00" layered, where "-" denotes the anionic S (Se) layer, "+" denotes the cationic Mo (Hf) layer, and "0" denotes the charge neutral GeO₂ layer. Due to the shielding effect of the "-+-" charge layer, the interface electric field only exists in GeO₂ (Supplementary Fig. 43). This field will pull down the energy bands of TMD relative to those of GeO₂, and also move the VBM state of TMD towards GeO₂. For pristine MoS₂ (MoSe₂), the VBM state resides on Mo. However, in the hetero-bilayer, part of this state shifts towards the S (Se) near GeO₂, leading to an emerging valence state at Gamma point, see the highest valence band at Gamma point in Fig. 4c, g, reducing the band gap. Conversely, for pristine HfS₂ (HfSe₂), the VBM state is on S (Se) and becomes more localized on the S (Se) near GeO₂ in the hetero-bilayer. This localization enhances the quantum constraint and decreases the hole energy level,

leading to an increase in the band gap. These results provide fundamental insights into the electronic properties of 2D GeO₂/TMD HBLs, and provide a reference for the design and fabrication of future 2D FETs.

To evaluate the impact of lattice mismatch on GeO₂ properties, we calculate the band gaps and static dielectric constants of the GeO₂ unit cell under different strains in the x-direction. Our calculated results, presented in Supplementary Table 33, reveal that the band gap of GeO₂ is perfectly preserved in the strain range we have considered, while the static dielectric constant of GeO₂ is highly sensitive to strain. However, the ultra-high κ value of GeO₂ can be maintained under a small strain. It is worth noting that the static dielectric constant of GeO₂ increases significantly under a 0.5% strain. Furthermore, we also compute the Poisson's ratio of GeO₂ on the HfSe₂ substrate. Our findings shown in Supplementary Fig. 44 indicate that for GeO₂ in the HBL, the pristine monolayer's NPR effect transforms into a biased-Poisson's ratio effect, with PPR appearing during compression, NPR appearing during stretching, and zero Poisson's ratio[47,48] appearing at near zero strain.

## Discussion
GeO₂ is the most promising 2D oxide dielectric, with an ultra-high κ value of 99, which is significantly higher than that of the highly regarded 2D dielectrics CaF₂ (κ ~ 6)[23] and β-Bi₂SeO₅ (κ ~ 22)[24]. GeO₂ monolayer can form a type-I heterostructure with MoSe₂ and HfSe₂, resulting in a band offset of about 1 eV that is sufficient to minimize leakage current caused by Schottky emission of carriers into the dielectric. This holds great promise for 2D field-effect transistors.

The dynamic stability of our most promising oxide dielectrics, $GeO_2$, $Fe_2O_3$, $WO_3$, and $RhO_2$, has been confirmed by phonon dispersions. Further investigation of the binding energy of hydrogen molecules on these monolayers demonstrates their surface inertness. AIMD simulations are conducted at room temperature, revealing that these monolayers can exist stably under ambient conditions, even in the presence of water, oxygen, or hydrogen. Additionally, the 2D anti-ferromagnetic material $Fe_2O_3$ predicted in this study holds great promise as a candidate for high-temperature applications, such as in energy storage and catalysis, due to its high Néel temperature and robust thermal stability. $GeO_2$, $WO_3$, and $RhO_2$ monolayers exhibit interesting out-of-plane NPR and biased NPR effects, demonstrating promising mechanical properties for applications in stretchable devices.

Recently, Zavabeti et al. have demonstrated that the alloying of elemental hafnium into a liquid gallium-based alloy results in the natural formation of a 2D interfacial oxide skin of $HfO_2$ ($\bar{1}11$) crystal plane on the metal surface when exposed to an oxygen-containing environment[19]. This is because the oxide that yields the greatest reduction in Gibbs free energy and the close-packed crystal plane with the lowest surface energy dominate the surface. Thus, when using the liquid bismuth-tin alloy (138 °C) as the solvent, many of our predicted metal oxides close-packed planes, such as $GeO_2$, $Fe_2O_3$, BaO, CaO, BeO, MgO, $SnO_2$, and $WO_3$, etc., can be accessed as 2D layers. Supplementary Table 34 presents the Gibbs free energy of formation for our predicted metal oxide monolayers, which is lower than that of $Bi_2O_3$ and $SnO_2$. The remaining predicted 2D oxides with higher Gibbs free energy than $Bi_2O_3$ and $SnO_2$, such as $RhO_2$, AgO, CuO, $PbO_2$, ZnO, CoO, and NiO, etc., can be synthesized using nanoparticles as intermediates. Yang Juan et al. have exhibited that 3D nanoparticles are initially formed from the molecular precursor solution and then transform into 2D nanosheets due to the thermodynamic driving force[59]. The close-packed plane with the lowest surface energy is expected to be dominant for the surface. We hope that our findings can stimulate additional experimental research into the 2D oxides we have identified, particularly in relation to $GeO_2$, $Fe_2O_3$, $WO_3$, and $RhO_2$.

In conclusion, our study proposes an approach for designing 2D oxides that are derived from nonlayered bulk oxides. This approach involves geometric screening of promising exfoliated planes and conducting vdW-DFT calculations of interplanar binding energy. According to the approach, we identify 51 easily/potentially exfoliable 2D stable oxides from 47 experimentally stable nonlayered bulk oxides. The exfoliation process of such strongly bonded oxides could have exfoliation energy as low as 0.39 J/m², comparable with the exfoliation energy of graphite. The approach can be extended to any nonlayered crystal, providing a flexible and cost-effective approach for mining 2D materials from largely untapped nonlayered materials space. For the predicted 2D stable oxides, we fully characterize their electronic ground states, static dielectric constants and mechanical properties, disclosing a wealth of 2D functional oxides, including auxetic high-$\kappa$ oxide dielectrics ($GeO_2$, $RhO_2$, and $WO_3$), AFM high-$\kappa$ oxide dielectric $Fe_2O_3$, valleytronic oxide BaO, and other auxetic monolayers ($OsO_2$, $PdO_2$, $PbO_2$, AgO, CuO). Our study opens the way to design pure-2D electronics based on nonlayered crystals, suggesting native oxide dielectrics for future 2D materials-based information technologies.

## Methods
### Calculation details
All calculations are performed within DFT[60] in the Vienna Ab initio Simulation Package (VASP)[61] using the projector-augmented wave (PAW) pseudopotentials[62] with the generalized gradient approximation GGA/PBE + U[63] exchange-correlation functional without considering the spin−orbit coupling effect. The $U$ values are energy corrections for the spurious self-interaction energy introduced by GGA. We use $U$ values for $d$ orbitals that are fully tested by fitting to experimental lattice constants and magnetic moments, as evidenced in Supplementary Tables 1, 2. Grimme's DFT-D3[64] scheme is adopted to describe the long-range vdW interactions to get an accurate estimation of $E_b$ and to harmonize the calculations with layered materials for comparison purposes. Our GGA + U calculations with Grimme's D3 dispersion correction can reasonably reproduce the $E_b$ for well-known layered materials calculated by DF2-C09 functional[17] and random phase approximation[22] (Supplementary Fig. 3). The dipole correction[65,66] is used to eliminate spurious electrostatic coupling between periodic copies in the $z$-direction for polar material. The hybrid function of HSE06[67] is used to calculate electronic structure and DOS. We employ the collinear approximation for the relevant DFT calculations of magnetic materials, except for the calculation of magnetic anisotropy in $Fe_2O_3$.

The plane-wave energy cutoff of 520 eV is used. The Brillouin zone is sampled by a Monkhorst−Pack K-point grid with a uniform spacing of 0.01 Å$^{-1}$ for DOS calculations, 0.02 Å$^{-1}$ for geometry optimization, dielectric, Poisson's ratio, elastic constant calculations, and phonon calculations. During exfoliation calculations, the numerical convergence of the structural relaxation is achieved with a tolerance of $10^{-5}$ eV in energy and 0.01 eV/Å in force. For 2D structures, a vacuum space of 15 Å is added to avoid the interactions between a layer and its replica. The 2D structures are fully optimized with a finer tolerance of $10^{-7}$ eV/Å in force. For static dielectric constant calculations, the energy convergence criteria is set to be $10^{-8}$ eV to obtain precise and reliable dielectric values. We calculate the force constants with density-functional perturbation theory (DFPT)[68] using VASP. The phonon dispersion is then obtained using Phonopy[69]. A supercell containing around 100 atoms for each 2D material with Gamma point K sampling in wave vector space is used for AIMD simulations. NVT ensemble simulations with a Nosé-Hoover thermostat at 300 K and a time step of 3 fs are carried out. The total simulation time is ~10 ps. For $E_{bind}$ [$H_2$] calculations, we place four hydrogen molecules on the surface of the supercell in a random manner. The adsorption density is ~$10^{14}$ molecules per square centimeter and the average distance between the molecules is ~10 Å. The calculation equation for $E_{bind}$ [$H_2$] is as follows[70,71],

$$E_{bind} = \frac{E_{tot}^{adsorbed\ 2D\ oxide} - E_{tot}^{pristine\ 2D\ oxide} - N \times E_{tot}^{isolated\ molecule}}{N} \quad (1)$$

where the numerator terms on the right-hand side represent the total energies of 2D oxides with $N$ adsorbed $H_2$ molecules, pristine 2D oxides, and $N$ isolated $H_2$ molecules, respectively. We also carry out K-point convergence tests (2D materials: $Fe_2O_3$, $BaO_2$, $OsO_2$, $GeO_2$, $SnO_2$, and $SrO_2$ as test cases) to show that our converged K-points are sufficient to predict the DFPT related data as shown in Supplementary Tables 25 and 26.

### Close-packed degree
For a truly close-packed plane made up of atoms of diameter $D$, the area per atom ($A_{cp}$) is given by $A_{cp} = \frac{\sqrt{3}}{2}D^2$. And for a given crystallographic plane ($h\ k\ l$), the area per atom ($A_{hkl}$) can be estimated from the following equation[72]:

$$A_{hkl} = \frac{V}{d_{hkl} \times |F_{hkl}|} \quad (2)$$

where $V$ is the volume of unit cell, $d_{hkl}$ is the spacing between adjacent plane ($h\ k\ l$), and $|F_{hkl}|$ denotes the structural factor for the crystallographic plane ($h\ k\ l$). It is easy to see from Eq. (2) that the larger the interplanar spacing $d_{hkl}$ and the structure factor, the smaller the value of $A_{hkl}$, i.e., the nearer the plane is to being close-packed. How close-packed of a specific crystallographic plane ($h\ k\ l$) in a given crystal can

be estimated by the ratio of $A_{cp}$ to $A_{hkl}$ and expressed as a percentage, i.e., $(A_{cp}/A_{hkl}) \times 100\%$.

## Comparison of crystal structure

We conduct a preliminary comparison of the lattice parameters and space group to determine whether the two crystals are identical. Then, we conduct a precise assessment of their identity by comparing the atomic positions of the two crystals. To accomplish this, we calculate the root-mean-square deviation (RMSD) of the atomic positions between the two structures. The RMSD is calculated as follows:

$$RMSD = \sqrt{\frac{1}{N} \sum_{i}^{natom} \left[ (x_i - x_i')^2 + (y_i - y_i')^2 + (z_i - z_i')^2 \right]} \quad (3)$$

where $i$ cycles through all atoms, $x_i$ and $x_i'$ are the $x$-coordinates of the $i^{th}$ atom in the two respective structures, respectively, and $y$ and $z$ are analogous. A low RMSD value indicates a high degree of similarity or identity between the two structures. Utilizing this approach, we identify eight identical crystal structures with RMSD values ranging from 0.0006 to 0.0109. For further details, please refer to Supplementary Table 21.

## Static dielectric tensor

The static dielectric constant ($\kappa$) includes both electronic and ionic contributions to the dielectric response. We employ density-functional perturbation theory to calculate the static dielectric tensor, from which we extract the in-plane (‖) and out-of-plane dielectric constants (⊥). The in-plane static dielectric constant is obtained by averaging the $x$ and $y$ components, namely, $\kappa_{\parallel} = (\kappa_x + \kappa_y)/2$, and the out-of-plane dielectric constant equals to the $z$ component.

Static dielectric tensor of 2D material computed by DFT contains the contribution of the material itself and the vacuum layer, due to the fact that the macroscopic electric field is exerted on the supercell containing the vacuum layer. Hence, to obtain material static dielectric constant, we need to remove the contribution of the vacuum layer using the following equations[29]:

$$\kappa_{\parallel}^{2D} = \frac{c}{t} \left( \kappa_{\parallel}^{sup} - 1 \right) + 1 \quad (4)$$

$$\kappa_{\perp}^{2D} = \left[ \frac{c}{t} \left( \frac{1}{\kappa_{\perp}^{sup}} - 1 \right) + 1 \right]^{-1} \quad (5)$$

where $c$ is the supercell height, and $t$ is the thickness of the monolayer. The thickness $t$ is estimated by the interlayer distance of the bilayer. To test the effect of long-range Coulomb interactions in the out-of-plane direction on the dielectric constant, we calculate the static dielectric constants of monolayers with vacuum size of 30 Å, which are in satisfactory agreement with that of 15 Å, see Supplementary Fig. 35. These results show that the environmental screening by periodic images does not affect the obtained dielectric constants.

## Poisson's ratio

When the lattice is strained in the $x$ or $y$-direction, the other in-plane lattice constant, the thickness of the monolayer and all the atoms in the system are fully relaxed. The strain along the $x$ or $y$-direction is defined as "Strain $x/y = (a - a_0)/a_0$", while the strain along the $z$-direction is defined as "Strain $z = (t - t_0)/t_0$", where $a_0$ and $t_0$ are the lattice constant and thickness of the freestanding monolayer, respectively, and $a$ and $t$ are the corresponding values at the strained states.

The orientation-dependent in-plane Poisson's ratio $\upsilon(\theta)$ of the 2D materials with orthogonal symmetry can be evaluated using the elastic stiffness constants ($C_{11}, C_{12}, C_{22}, C_{66}$) with the following equation[73]:

$$\upsilon(\theta) = \frac{C_{12}\sin^4\theta - B\sin^2\theta\cos^2\theta + C_{12}\cos^4\theta}{C_{11}\sin^4\theta + A\sin^2\theta\cos^2\theta + C_{22}\cos^4\theta} \quad (6)$$

where, $A = (C_{11}C_{22} - C_{12}^2)/C_{66} - 2C_{12}$, $B = C_{11} + C_{22} - (C_{11}C_{22} - C_{12}^2)/C_{66}$, and $\theta$ denotes the angle of the applied strain with respect to the $x$-direction.

## Data availability

The Interplanar binding energy, interplanar spacing, the close-packed degree, crystal structures, U values, magnetic moments, phonon dispersion, AIMD simulations, hydrogen binding energy, static dielectric constant, band gap, vdW gap, and elastic constant data generated in this study are provided in the Supplementary Information file.

## Code availability

The codes that are necessary to reproduce the findings of this study are available from the corresponding author upon request. All DFT calculations were performed by using the Vienna ab initio simulation package (VASP).

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

## Acknowledgements

This work is supported by Natural Science Foundation of Guangdong Province, China (2022A1515011990, 2023A1515030086), National Natural Science Foundation of China (Grant Nos. 11804230, 11774239, 61827815), National Key R&D Program of China (2019YFB2204500), and Shenzhen Science and Technology Innovation Commission (Grant Nos. JCYJ20220531102601004, KQTD20180412181422399, JCYJ20180507181858539).

## Author contributions

Y.H., P.H., and X.Z. conceived the project. Y.H. and X.Z. proposed using geometric criteria to identify promising exfoliable crystallographic planes from nonlayered crystals. Y.H. developed the code and carried out the majority of calculations. Z.M. conducted the ab initio molecular dynamics calculations. Y.H., P.H., and X.Z. analyzed the obtained results and wrote the paper. J.J., P.Z., Z.M., F.G., D.L., Z.Q. and X.Z. contribute to the discussion of the manuscript.

## Competing interests

The authors declare no competing interests.
