## [Peer Review File · Nature Communications]

Prediction of nonlayered oxide monolayers as flexible high- κ dielectrics with negative Poisson's ratiosEditorial Note: Parts of this Peer Review File have been redacted as indicated to remove third-party material where no permission to publish could be obtained.

REVIEWER COMMENTS

Reviewer #1 (Remarks to the Author):

The idea behind the paper is very interesting, the authors investigate which oxides are potentially exfoliable and investigate the properties of these potentially exfoliable oxides in their monolayer form. The paper contains a lot of data but there are a number of issues that prevent it from being published in its current form. If the following are addressed, I believe publication in Nature Communications would be appropriate.

1) The materials that are identified based on the search have important practical issues and will unlikely see practical applications

- GeO₂ which is listed in the abstract and throughout the paper has historically been left behind in favor of SiO₂ because of water solubility. How much of an issue would this be to demonstrate and use the material experimentally?

- Fe₂O₃ is mentioned in the conclusion but it is unclear how the anti-ferromagnetic properties would be used since it forms a hexagonal lattice and there is unlikely to be long-range anti-ferromagnetic order

The authors should provide a perspective on the most promising materials and perhaps narrow them down based on experimental considerations

2) Basic DFT computational details are missing. What k-point grid is used? What grid is used for the phonons in the DFPT? The convergence threshold on the electronic properties also seems too low to get accurate dielectric constants

3) Some phonon spectra exhibit negative spectra, the instability should be mentioned in the paper, e.g. supplementary table 3

4) The methodology to calculate Fig. 3 is not explained. How is the strain in the z-direction calculated? How much of this property remains when the material is on a substrate or encapsulated?

5) The authors mention negative Poisson's ratio but I see positive Poisson's ratio in Fig. 3e-f

6) Use different symbols to indicate the different properties in Fig. 1 instead of just circles.

Reviewer #2 (Remarks to the Author):

The authors discuss an approach to extract 2D materials out of parent ones where the parent material is not layered. They focus in particular on oxides with the aim of identifying high-k dielectrics for e.g. heterostructure applications in nano electronics.

The approach is based on gradually expanding the lattice parameter along one direction and then computing the binding energy while relaxing atoms (and where two different qualitative behaviours are identified, depending on whether bonds reconstruct).

The approach is different from papers in the literature that e.g. start from layered parent compounds (e.g. Nature Nanotech 13, 246–252 (2018)) or combinatorial lattice decoration (e.g. J. Phys. Chem. C 119, 13169–13183 (2015)).

The identified layers are then investigated in detail identifying candidates with interesting mechanical and dielectric properties.

While the paper provides some potentially interesting results, I have some questions for the authors both to better understand the novelty of the results, and to understand the realistic possibility of realising such materials, before I can recommend it for publication.

- Can the authors describe in more detail the exact algorithm used to extract the 2D materials? For instance:
 - how many steps are considered for the c lattice parameter and with which step size
 - is a relaxation is performed at every step? Is the relaxation for everything (in-plane cell+atoms), or only atoms, or with some additional constraints, ... In particular, if the in-plane cell is relaxed or not can significantly affect the results.
 - is this approach novel or it was already used in the literature?
 - can the authors add a figure at least for the few most relevant materials, similar to Figs. 1a and 1d, but with actual numbers and units on the axes to have a more quantitative understanding of the approach?

- How confident can we be that the layers can actually be exfoliated? My doubt is the following. Even if the binding energy is low, do we know that, in plane, the binding energy is still high? If it is small also in plane, the layer would "break" when trying to exfoliate it. The authors make some arguments based on the packing ratio, that intuitively might make sense. However, looking at e.g. the bulk structure of GeO₂ and just looking at the bond lengths, it's not obvious that the in-plane bonds remain very strong. This is of course one of the arguments of the authors. Can they test this? E.g. trying to "break apart" some of the best candidates in-plane, to see if they are still strongly bounded, e.g. computing the binding energy also in plane once the 2D layer is formed? Or use some other approach to assess their actual stability (e.g. some short molecular dynamics at room temperature).

Some additional comments:

- when creating heterostructures, lattice-matching can be very important. Did the authors consider lattice-matching considerations in their analysis of heterostructures? If so, can this be better commented? If not, can the authors discuss the strain required to lattice-match the layers, and check the changes in properties (e.g. in the 2D materials alone) due to this strain?

- The C2DB database (<https://cmrdb.fysik.dtu.dk/c2db>) provides some GeO₂ structures. Could the authors compare the structures they identify (not only GeO₂, also the other candidates) with those existing in the C2DB and assess if they are the same or they are different?

- Authors discuss the phonon dispersion of their selected materials. Are materials filtered based on whether the phonons are real and positive? As they discuss, most of them are, but a few have imaginary phonons ($\omega^2 < 0$), e.g. AgO (101), NiO (011), K₂O (110), ... Are they discarded? If not, I suggest to use an approach such as the one of Togo, Tanaka, Phys. Rev. B 87, 184104 (2013):

<https://journals.aps.org/prb/abstract/10.1103/PhysRevB.87.184104> - it is relatively easy to use and can provide stable structures (possibly in a larger supercell) starting from the unstable ones. Those with unstable phonons are unstable and therefore not the final structures (and this can affect e.g. their mechanical properties).

- at lines 253-256, authors say "the product of m_{eff} and Φ_b is roughly proportional to

E_g : can they justify this? Also, then they replace $(m_{\text{eff}} \cdot \Phi_b)^{1/2}$ with E_g (without square root), so the two sentences are not consistent. Please explain in more detail.

- Lines 272-273: authors say that the Ge electronic states in the valence and conduction are mainly from Ge in the edge layer. However, From Fig. 2c, for the first peak in the valence bands, the two contributions are essentially the same (so 50% from edge and 50% from the middle layer). Please fix the discussion.

- Line 275: authors speak of a "hybridisation between unusually large number of O p valence bands and...". How can we say that it's "unusually large"? Can they e.g. show what happens in the bulk and that a smaller number of states hybridise, or some other justification?

- Which 2D layers have a dipole along z? For those, do the authors use a correct screening of the long-range electrostatic field? This can be very important to remove spurious effects due to the PBC.

- Do the authors use a correct treatment of the dielectric screening for 2D materials? This can significantly change the relevant dielectric properties of the material. See e.g. Sohier et al, Nano Lett. 17, 3758 (2017), <https://pubs.acs.org/doi/pdf/10.1021/acs.nanolett.7b01090> ; the effects can be very important, with a different treatment of screening and the disappearance (in 2D) of the well-known LO-TO phonon splitting (that exists instead in 3D). In particular: is ϵ_{2D} of Suppl. Figures 32, 33 defined in the same sense of ϵ_{2D} in the citation above (Sohier 2017, Eq. (3))?

- Authors say that GeO₂ forms Type-II heterostructures with HfS₂. However, the valence bands are essentially aligned, so it's ambiguous if it's type-I or type-II?

- Line 394: authors say that the gap reduction is mostly caused by the drop of the CBM in MoS₂. However, from the band structure of Fig. 4c (and comparing with the band structure of pristine MoS₂ in the Supplementary), a big change is given by the "new" valence band at Gamma that has a much higher energy. So I am not sure one can say that the change is mostly due to the drop of the conduction band.

- Lines 394-399: the discussion is very interesting. However I think it should be better clarified (it's not so clear from the text, one has to think about it) that atoms on the edge are those mostly affected by the heterostructuring w.r.t. atoms in the middle of the 2D layer, and therefore the effect in the materials discussed is different depending on the projection of near-gap states on the edge or middle atoms.

- line 426 (methods): authors used DFT+U. Which values of U did they use, for which atoms,

and how did they choose this value(s)?

- In the supplementary (workflow section), they say that they added oxide dielectrics TiO₂, ZrO₂, HfO₂. Why? They were not in the original database? Why they had to be added by hand?
- Can the authors comment on the structure of Ag₂O (111) (Suppl. Figure 29) - it seems that there are isolated atoms, or this is just what it looks like from the visualisation? What is the binding energy for removing those atoms? (i.e. are they really part of the 2D layers?)

In addition, I'd like to recommend some improvements on the language to the authors. Here just a few examples of sentences that do not sound correct in English (there are quite a few more):

- Line 66: geometrically screening potentially exfoliable crystallographic plane -> this sentence is incorrect from a grammatical point of view and not very clear
- Line 95: criteria -> criterion
- Line 104: is independently -> is independent
- Line 105: to any nonlayered crystals -> to any nonlayered crystal
- Line 300: if a stretched/compressed [...] is applied -> if a stretch/compression [...] is applied
- Line 303: possess [...] higher shear resistant -> resistant is not the correct word here!
- ...

Some final comments:

- the authors use interchangeably k and ϵ . Do they represent the same quantity, and if so can the same symbol be used? or can the difference be clarified?
- some supplementary tables and figures are not cited (e.g. it would be useful to cite Suppl. Fig. 37 when discussing the anomalous auxetic effect of WO₃)
- line 202 of the PDF: they mention MnO and 21 similar structures: shouldn't it be "19 similar structures"?
- in Fig. 3e and 3f, does the distance from the center of the colored bars represent the same value (Poisson ratio) as the color bar? Can the author clarify this better in the caption? In

addition, if this is the case, I don't understand why at zero angle, in Fig. 3f, the curve for WO₃ is red (while for AgO is still yellow) but the one of WO₃ is inside (at smaller distance from the center).

- supplementary table 2: add units for d_{hkl} , and add link to the section defining the % close-packed. Also, it spans a very large range, 8% to 100%. Can they show two extreme examples (8% and 100% for instance) and comment on this large range?

- I am not sure if this is just an issue of the conversion to PDF: but can the authors provide higher-resolution images? e.g. the spin arrows in Suppl. Fig. 39 (panels b and c) are not really clear with the resolution of the PDF file I have.

Reviewer #3 (Remarks to the Author):

The manuscript presents a strategic discovery and design of oxide monolayers by exfoliating nonlayered oxides with potentially low exfoliation energy, resulting in the identification of 61 potentially exfoliable and dynamically stable 2D oxides. The predicted 2D oxide families are expected to possess promising dielectric, mechanical, magnetic, and optoelectronic properties for technical applications, which could greatly diversify existing 2D materials in terms of material classes and functionalities if they are successfully synthesized experimentally. Overall, given the potential impact of this work in the field of functional 2D materials, I recommend the publication of the manuscript, provided that the following comments are addressed properly.

1. One concern regarding the predictions of oxide monolayers from nonlayered oxides is the potential lack of surface inertness that could compromise their stability in practical applications. The authors should address this concern by providing examples demonstrating that the surfaces of these monolayers are stable enough against molecule absorption. Specifically, it would be helpful to know the typical hydrogen binding energy on these monolayers to assess their stability.

2. Many of the newly predicted oxide monolayers exhibit out-of-plane negative Poisson's ratio but positive in-plane Poisson's ratio. It would be helpful for the authors to comment on

the difference between the oxide monolayers and those 2D materials that have been predicted to have in-plane negative Poisson's ratio, such as the ones reported by J. Pan et al. in *npj Computational Materials* 6, 154 (2020) and L. Yu et al. in *Nature Communications* 8, 15224 (2017). This comparison would provide valuable context and help readers better understand the significance of the current findings.

3. Another question related to the manuscript is the effectiveness of the 2D structure discovery strategy employed. While the use of close-packed planes in nonlayered structures is an intelligent way to reduce the computational cost of the structure discovery process, it is possible that the final monolayer structures identified may already exist in many layered materials. In this case, there may be significant overlap between the final structures obtained from either layered or nonlayered initial structures. The authors should address this concern by checking the fraction of identified oxide monolayer structures that have already been included in other 2D material databases such as the Computational 2D Materials Database (C2DB), which is constructed based on layered bulk materials and elemental substitution. Such an analysis would help demonstrate the novelty of the discovered oxide monolayers and provide insights into the effectiveness of the discovery strategy employed.

4. The authors stated that only a few of the predicted oxide monolayers have been experimentally studied. Therefore, it would be helpful if the authors could provide some comments and guidance on how these oxide monolayers can be synthesized through viable routes. Such information would be useful for researchers interested in synthesizing these materials and could help accelerate the experimental validation of the predicted properties.

I hope these comments will be helpful for the authors in improving the manuscript.

Comments and Author reply

Reviewer #1 (Remarks to the Author):

Comment: *The idea behind the paper is very interesting, the authors investigate which oxides are potentially exfoliable and investigates the properties of these potentially exfoliable oxides in their monolayer form. The paper contains a lot of data but there are a number of issues that prevent it from being published in its current form. If the following are addressed, I believe publication in Nature Communications would be appropriate.*

Reply: We sincerely thank the referee for carefully reviewing our work and for providing a positive view of our work.

Comment 1.1.1: *1) The materials that are identified based on the search have important practical issues and will unlikely see practical applications
- GeO₂ which is listed in the abstract and throughout the paper has historically been left behind in favor of SiO₂ because of water solubility. How much of an issue would this be to demonstrate and use the material experimentally?*

Reply: We thank the referee for raising the issue of the water solubility of GeO₂. Firstly, to assess the actual stability of 2D GeO₂ when exposed to moisture, we conducted *ab initio* molecular dynamics simulations at room temperature under a H₂O, O₂, or H₂ containing environment. The results shown in Figure R1 demonstrate that although some atomic coordinates had changed slightly, the atomic structure remained stable. This indicates that our predicted 2D GeO₂ can withstand moisture at finite temperature under ambient conditions. Secondly, it is worth noting that the single-crystalline ionic crystal calcium fluoride (CaF₂) has been utilized as a dielectric layer in back-gated 2D devices [Nature Electronics **2**, 230–235 (2019)], despite being capable of dissolving in water to some extent (0.0016 g in 100 g of H₂O at 25 °C) [CRC Handbook of Chemistry and Physics], suggesting that low water solubility did not hinder the experimental use of CaF₂. Lastly, the silica bilayer is currently the thinnest attainable layer with the stoichiometry SiO₂. However, silica bilayer consists of amorphous and crystalline domains [PRL **109**, 106101 (2012)]. The application of polycrystalline materials is probably limited by defects associated with grain boundaries, and by the interfacial roughness arising from potentially faceted interfaces. What's worse, the already low dielectric constant of bulk SiO₂ is further reduced when it becomes 2D SiO₂ ($k < 2.5$) [Materials Science and Engineering C **27**, 1145–1148 (2007)]. It has been reported that if the SiO₂ layer becomes thinner than ~1 nm, the leakage current due to the quantum tunneling effect begins to dominate, which causes serious problems in power consumption and device performance [Nature **406**, 1032–1038 (2000)]. In sharp contrast, our predicted 2D GeO₂ exhibits an ultra-high k value of ~99,

increasing the physical thickness and significantly reducing direct tunneling, holding great potential to overcome the aforementioned dilemma of SiO₂ film.

Fig. R1 AIMD simulation for 10 ps at 300K for the 2D dielectric oxide GeO₂ with adsorption of ambient molecules (H₂O, H₂, or O₂) from left to right, indicating stable surface.

Comment 1.1.2: - *Fe₂O₃ is mentioned in the conclusion but is unclear how the anti-ferromagnetic properties would be used since it forms a hexagonal lattice and there is unlikely to be long-range anti-ferromagnetic order*

Reply: We thank the referee for bringing up the issue of long-range anti-ferromagnetic order in Fe₂O₃ and for considering the possibility of utilizing its anti-ferromagnetic properties. In the current Supplementary Table 4 (previous Table 3), we have presented the magnetic moment (5 μ_B) and energy difference between ferromagnetic (FM) and anti-ferromagnetic (AFM) configurations ΔE (0.2593 eV/Fe-atom) of our predicted 2D Fe₂O₃ obtained from first-principles calculations. The ground-state magnetic configuration is determined by comparing the energy of FM and all possible AFM ordering of on-site spins in the cell containing four iron atoms. Within mean-field theory, the transition temperature of the Ising model is $T_N = \alpha \Delta E (1 + S^{-1}) / k_B$ [$\alpha \sim 0.14$, Physical Review B **50**, 5041-5054 (1994)], where S is the total spin moment, and k_B is the Boltzmann constant. By utilizing our values of ΔE in conjunction with the magnetic moment, we can estimate that the Néel temperature (T_N) of our predicted 2D Fe₂O₃ is equal to 506.5 K, indicating a stable antiferromagnetic state at/above room temperature. The high T_N and robust thermal stability (see Figure R2) of 2D Fe₂O₃ makes it a promising candidate for applications in high-temperature environments, such as in catalysis and energy storage. The 2D nature of Fe₂O₃ also offers a large surface area, which can significantly enhance its catalytic activity. A type-II Fe₂O₃/MoS₂ heterostructure with the conduction band minimum (valence band maximum) located at Fe₂O₃ (MoS₂) is found by our first-principles calculations, which presents exciting opportunities for optoelectronics and photocatalysis [Nature Communications **6**, 7311 (2015), Nature Nanotechnology **11**, 42–46 (2016)].

Fig. R2 AIMD simulation for 10 ps at 300K for the 2D AFM dielectric oxide Fe_2O_3 (101), indicating that the predicted 2D Fe_2O_3 can exist quite stably at room temperature.

Finally, we note that one important contribution of our work is to propose a new approach to design 2D oxides from nonlayered bulk oxides, including 2D magnetic oxides. This approach is independent of chemical composition, bonding type and symmetry, and hence can be extended to any nonlayered crystal, providing a flexible and cost-effective tool for mining 2D nonmagnetic as well as magnetic materials from largely untapped nonlayered bulk materials space.

Comment 1.1.3: *-The authors should provide a perspective on the most promising materials and perhaps narrow them down based on experimental considerations.*

Reply: We would like to express our gratitude to the referee for raising the important comments mentioned above and for the valuable recommendation that we provide a perspective on the most promising materials and narrow down the selection based on experimental considerations. To fully address these very important comments, we have revised our manuscript in the following way (the blue fonts are newly added or revised contents),

- 1) We expanded the list of promising 2D high- k dielectrics from " GeO_2 and Fe_2O_3 " to " GeO_2 , Fe_2O_3 , WO_3 , and RhO_2 ". Both WO_3 (with $k \sim 30$) and RhO_2 (with $k \sim 62$) have higher dielectric constants than that of HfO_2 , and exhibit interesting out-of-plane NPR and biased-NPR effects, demonstrating promising mechanical properties for the application in stretchable devices. Furthermore, RhO_2 and WO_3 are insoluble in water, with RhO_2 specifically unable to be dissolved in acids or bases [CRC Handbook of Chemistry and Physics].

“In addition to the GeO_2 monolayer, we have also identified several noteworthy candidates, including Fe_2O_3 (020)/(101), WO_3 (110), and RhO_2 (110). Fe_2O_3 , WO_3 , and RhO_2 have higher dielectric constants than that of HfO_2 , with k values of approximately 24, 30, and 62, respectively. The PDOSs of the Fe_2O_3 , WO_3 , and RhO_2 monolayers shown in Fig. 2(e-g) demonstrate that the O p valence bands and corresponding transition metal (Fe, W, Rh) d conduction bands exhibit strong cross-gap hybridization, which leads to the large static dielectric constants.” (P. 12)

- 2) We have added the following contents to illustrate *how the anti-ferromagnetic properties would be used*, as follows:

“Additionally, the 2D AFM dielectric oxide Fe_2O_3 exhibits a large magnetic moment of $5 \mu\text{B}$ per Fe atom and an exchange energy of $\Delta E = 0.2593 \text{ eV/Fe-atom}$ (see Supplementary Table 4). Within mean-field theory, the transition temperature of the Ising model is³⁸ $T_N = \alpha\Delta E(1 + S^{-1})/k_B$ with $\alpha \sim 0.14$, where S is the total spin moment, and k_B is the Boltzmann constant. By substituting our values of ΔE and the magnetic moment, we can estimate that the predicted 2D Fe_2O_3 has a Néel temperature (T_N) of 506.5 K, indicating a stable antiferromagnetic state at or above room temperature. The high T_N and benign thermal stability of 2D Fe_2O_3 make it a promising candidate for various high-temperature applications, including catalysis and energy storage. Also, the 2D nature of Fe_2O_3 offers a large surface area, which could significantly enhance its catalytic activity. Our first-principles calculations also reveal a type-II $\text{Fe}_2\text{O}_3/\text{MoS}_2$ heterostructure, in which the conduction band minimum and valence band maximum are located at Fe_2O_3 and MoS_2 , respectively (see Supplementary Fig. 45). This discovery presents exciting opportunities for optoelectronics and photocatalysis^{39,40}.” (P. 12)

- 3) We added Supplementary Figure 45, which offers the electronic band structure of $\text{Fe}_2\text{O}_3/\text{MoS}_2$ heterostructure, as in Fig. R3 shown below.

Fig. R3. Projected band structures of the $\text{Fe}_2\text{O}_3/\text{MoS}_2$ heterostructure with type-II band alignment, i.e., the conduction band minimum (valence band maximum) is located at Fe_2O_3 (MoS_2). A 1×2 supercell of MoS_2 is used to match a 1×1 Fe_2O_3 cell. The strains applied to the Fe_2O_3 along the x and y directions are 3.5% and 1.9%, respectively.

- 4) We presented a perspective on the most promising 2D materials based on their physical properties and actual stability.

“ GeO_2 is the most promising 2D oxide dielectric, with an ultra-high k value of 99, which is significantly higher than that of the highly regarded 2D dielectrics CaF_2 ($k \sim 6$)²³ and $\beta\text{-Bi}_2\text{SeO}_5$ ($k \sim 22$)²⁴. GeO_2 monolayer can form a type-I heterostructure with MoSe_2 and HfSe_2 , resulting in a band offset of about 1 eV that is sufficient to minimize leakage current caused by Schottky emission of carriers into the dielectric.

This holds great promise for 2D field-effect transistors.

The dynamic stability of our most promising oxide dielectrics, GeO₂, Fe₂O₃, WO₃, and RhO₂, has been confirmed by phonon dispersions. Further investigation of the binding energy of hydrogen molecule on these monolayers demonstrates their surface inertness. AIMD simulations are conducted at room temperature, revealing that these monolayers can exist stably under ambient conditions, even in the presence of water, oxygen, or hydrogen. Additionally, the 2D antiferromagnetic material Fe₂O₃ predicted in this study holds great promise as a candidate for high-temperature applications, such as in energy storage and catalysis, due to its high Néel temperature and robust thermal stability. GeO₂, WO₃, and RhO₂ monolayers exhibit interesting out-of-plane NPR and biased-NPR effects, demonstrating promising mechanical properties for applications in stretchable devices.” (P. 18)

- 5) We discussed two synthesized methods that have the potential to be utilized for synthesizing all of the oxide monolayers that we predict. The first method is the liquid metal-assisted exfoliation technique proposed by A. Zavabeti et al. in 2017 [Science **358**, 332-335 (2017)]. The second technique involves the formation of 2D transition metal oxide nanosheets with nanoparticles as intermediates, which was proposed by Yang Juan et al. in 2019 [Nature Materials **18**, 970-976 (2019)]. Following this, we have further narrowed down the list of materials based on the feasibility of experimental synthesis.

“Recently, Zavabeti et al. have demonstrated that the alloying of elemental hafnium into a liquid gallium-based alloy results in the natural formation of a 2D interfacial oxide skin of HfO₂ ($\bar{1}11$) crystal plane on the metal surface when exposed to an oxygen-containing environment¹⁹. This is because the oxide that yields the greatest reduction in Gibbs free energy and the close-packed crystal plane with the lowest surface energy dominate the surface. Thus, when using the liquid bismuth-tin alloy (138°C) as the solvent, many of our predicted metal oxides close-packed planes, such as GeO₂, Fe₂O₃, BaO, CaO, BeO, MgO, SnO₂ and WO₃, etc., can be accessed as 2D layers. Supplementary Table 33 presents the Gibbs free energy of formation for our predicted metal oxide monolayers, which is lower than that of Bi₂O₃ and SnO₂. The remaining predicted 2D oxides with higher Gibbs free energy than Bi₂O₃ and SnO₂, such as RhO₂, AgO, CuO, PbO₂, ZnO, CoO, and NiO, etc., can be synthesized using nanoparticles as intermediates. Yang Juan et al. have exhibited that 3D nanoparticles are initially formed from the molecular precursor solution and then transform into 2D nanosheets due to the thermodynamic driving force⁵⁶. The close-packed plane with the lowest surface energy is expected to be dominant for the surface. We hope that our findings can stimulate additional experimental research into the 2D oxides we have identified, particularly in relation to GeO₂, Fe₂O₃, WO₃, and RhO₂.” (P. 18-19)

Comment 1.2: 2) *Basic DFT computational details are missing. What k-point grid is used? What grid is used for the phonons in the DFPT? The convergence threshold on*

the electronic properties also seems too low to get accurate dielectric constants.

Reply: We thank the referee for the careful reading of our paper. We regret for missing some basic calculation details in the Methods section of our previous manuscript. We have described the K-point grid and DFPT related calculations in details in the **Methods** section in the revised manuscript, as shown below (the blue fonts are newly added or revised contents):

“The plane-wave energy cutoff of 520 eV is used. The Brillouin zone is sampled by a Monkhorst–Pack K-point grid with a uniform spacing of 0.01 Å⁻¹ for DOS calculations, 0.02 Å⁻¹ for geometry optimization, dielectric, Poisson’s ratio, elastic constant calculations, and phonon calculations. During exfoliation calculations, the numerical convergence of the structural relaxation is achieved with a tolerance of 10⁻⁵ eV in energy and 0.01 eV/Å in force. For 2D structures, a vacuum space of 15 Å is added to avoid the interactions between a layer and its replica. The 2D structures are fully optimized with a finer tolerance of 10⁻⁷ eV/Å in force. For static dielectric constant calculations, the energy convergence criteria is set to be 10⁻⁸ eV to obtain precise and reliable dielectric values. We calculate the force constants with density functional perturbation theory (DFPT)⁶⁵ using VASP. The phonon dispersion is then obtained using Phonopy⁶⁶. A supercell containing around 100 atoms for each 2D material with Gamma point K sampling in wave vector space is used for AIMD simulations. NVT ensemble simulations with a Nosé-Hoover thermostat at 300 K and a time step of 3 fs are carried out. The total simulation time is ~10 ps. For E_{bind} [H₂] calculations, we place four hydrogen molecules on the surface of the supercell in a random manner. The adsorption density is ~10¹⁴ molecules per square centimeter and the average distance between the molecules is ~10 Å. The calculation equation for E_{bind} [H₂] is as follows⁶⁷⁻⁶⁸,

$$E_{bind} = \frac{E_{tot}^{adsorbed\ 2D\ oxide} - E_{tot}^{pristine\ 2D\ oxide} - N \times E_{tot}^{isolated\ molecule}}{N} \quad (1)$$

where the numerator terms on the right-hand side represent the total energies of 2D oxides with N adsorbed H₂ molecules, pristine 2D oxides, and N isolated H₂ molecules, respectively. We also carry out K-point convergence tests (2D materials: Fe₂O₃, BaO₂, OsO₂, GeO₂, SnO₂, and SrO₂ as test cases) to show that our converged K-points are sufficient to predict the DFPT related data as shown in Supplementary Tables 24, 25.” (P. 19-21)

We have added Supplementary Tables 24, 25 (presented as Tables R1 and R2) to illustrate the K-point convergence tests for the phonon and dielectric calculations. It is evident that that the K-points utilized in our calculations are adequate for predicting the DFPT related data.

Table R1. K-point dependence of maximum and minimum optical phonon modes for Fe₂O₃ (101), OsO₂ (110), and BaO₂ (001) monolayers.

system	Supercell	KP-spacing (Å ⁻¹)	KP-mesh	Max mode (THz)	Min mode (THz)

BaO ₂ (001)	4 × 4 × 1	0.04	7 × 7 × 1	24.2406	4.0623
		0.02	13 × 13 × 1	24.2412	4.0721
		0.01	27 × 27 × 1	24.2410	4.0722
Fe ₂ O ₃ (101)	2 × 3 × 1	0.04	5 × 8 × 1	19.2607	2.6043
		0.02	9 × 17 × 1	19.2598	2.6035
		0.01	19 × 34 × 1	19.2599	2.6036
OsO ₂ (110)	2 × 2 × 1	0.04	4 × 4 × 1	22.4131	2.2383
		0.02	8 × 8 × 1	22.4132	2.2377
		0.01	17 × 15 × 1	22.4132	2.2376

Table R2. K-point dependence of average dielectric constant for BaO₂ (001), Fe₂O₃ (101), GeO₂ (011), SnO₂ (220), and SrO₂ (001) monolayers. The standard conventional cell for each material is used for the static dielectric constants calculation.

system	KP-spacing (Å ⁻¹)	KP-mesh	k
BaO ₂ (001)	0.02	13 × 13 × 1	13.70
	0.01	27 × 27 × 1	13.79
Fe ₂ O ₃ (101)	0.02	9 × 17 × 1	23.96
	0.01	19 × 34 × 1	24.00
GeO ₂ (101)	0.02	5 × 12 × 1	98.98
	0.01	9 × 25 × 1	97.50
SnO ₂ (220)	0.02	12 × 16 × 1	11.61
	0.01	25 × 31 × 1	11.59
SrO ₂ (001)	0.02	14 × 13 × 1	16.62
	0.01	28 × 27 × 1	17.21

Comment 1.3: 3) *Some phonon spectra exhibit negative spectra, the instability should be mentioned in the paper, e.g. supplementary table 3*

Reply: We thank the referee for the suggestion. We have supplemented the following contents to the section “**Nonlayered Oxides Exfoliation**” in the revised manuscript, to provide a description of phonon dispersions with negative spectra (the blue fonts are newly added or revised contents).

“The computed phonon dispersions, shown in Supplementary Figs. 5-30, reveal that most of the oxide monolayers presented non-imaginary frequency phonon dispersions, indicating dynamic stability, while the remaining about 21% of the monolayers exhibit negative phonon spectra, predicted to be unstable. For these unstable monolayers, one could search for their stable or metastable structures by investigating the phase-transition pathway²⁸. However, due to workload constraints, we exclude the unstable monolayers from further study in this work.” (P. 7)

Comment 1.4: 4) *The methodology to calculate Fig. 3 is not explained. How is the strain in the z-direction calculated? How much of this property remains when the material is on a substrate or encapsulated?*

Reply: We thank the referee for the questions for Fig. 3 in our main text. We define the strain along the out-of-plane z -direction as "Strain $Z = (t - t_0)/t_0$ ", and the strain along the in-plane x/y direction as "Strain $X/Y = (a - a_0)/a_0$ ". Here, t_0 and a_0 are the thickness and lattice constant of the freestanding monolayer, respectively, while a and t represent the corresponding values at the strained states. When the lattice undergoes a specific uniaxial loading condition, the other in-plane lattice constant, the thickness of the monolayer, and all the atoms in the system are fully relaxed.

In our revised manuscript, we have added a new subsection titled "**Poisson's Ratio**" in the Methods section (as shown below) to elaborate on the calculation methodology employed to derive the results presented in Fig. 3 of manuscript. Additionally, we have added the sentence "The relevant calculation details are provided in the Methods section" in page 12 of the revised main text to assist readers in locating the pertinent information.

“Poisson's ratio. When the lattice is strained in the x or y direction, the other in-plane lattice constant, the thickness of monolayer and all the atoms in the system are fully relaxed. The strain along the x or y direction is defined as “Strain $X/Y = (a-a_0)/a_0$ ”, while the strain along the z -direction is defined as “Strain $Z = (t-t_0)/t_0$ ”, where a_0 and t_0 are the lattice constant and thickness of the freestanding monolayer, respectively, and a and t are the corresponding values at the strained states.

The orientation-dependent in-plane Poisson's ratio $\nu(\theta)$ of the 2D materials with orthogonal symmetry can be evaluated using the elastic stiffness constants ($C_{11}, C_{12}, C_{22}, C_{66}$) with the following equation⁷⁰:

$$\nu(\theta) = \frac{C_{12}\sin^4\theta - B\sin^2\theta\cos^2\theta + C_{12}\cos^4\theta}{C_{11}\sin^4\theta + A\sin^2\theta\cos^2\theta + C_{22}\cos^4\theta} \quad (6)$$

where, $A = (C_{11}C_{22} - C_{12}^2)/C_{66} - 2C_{12}$, $B = C_{11} + C_{22} - (C_{11}C_{22} - C_{12}^2)/C_{66}$, and θ denotes the angle of the applied strain with respect to the x -direction.” (P. 22-23)

We thank the referee for posing a thought-provoking question regarding the preservation of the auxetic effect of the monolayers when they are on a substrate or encapsulated. To address this inquiry, we take monolayers of GeO_2 and PdO_2 as examples to examine their NPR on substrates (graphene and HfSe_2). The relaxed lattice constants are given in Table R3. To construct heterobilayer (HBL), a 3×2 supercell of PdO_2 monolayer is used to match a 2×5 supercell of graphene substrate, and a 3×1 supercell of GeO_2 monolayer is used to match a 2×3 supercell of HfSe_2 substrate. The strains applied to the monolayers of PdO_2 and GeO_2 are approximately 2.7% (x -direction), 2.0% (y -direction), and 2.6% (x -direction), 0.2% (y -direction), respectively.

Table R3. The relaxed lattice parameters for WO_3 , PdO_2 , GeO_2 , graphene and HfSe_2 monolayers.

TMDs	a (Å)	b (Å)	γ (°)
PdO ₂	3.01	6.40	90
graphene	4.27	2.46	90
GeO ₂	4.01	10.9	90
HfSe ₂	6.33	3.65	90

Next, we calculate the out-of-plane Poisson's ratio of PdO₂, and GeO₂ in their respective HBLs under a uniaxial load in either the x -direction or y -direction, as shown in Fig. R4. The calculations reveal significant differences under the influence of the substrate on the monolayer. In the case of PdO₂, the NPR effect is fully preserved in HBL, despite a slightly smaller NPR value of -0.086 compared to the pristine PdO₂ monolayer with an NPR value of -0.172. In sharp contrast, for GeO₂ in the HBL, the pristine monolayer's NPR effect transforms into a biased-Poisson's ratio effect, with PPR appearing during compression and NPR appearing during stretching. Remarkably, at near zero strain, the Poisson's ratio for GeO₂ in HBL becomes zero, analogous to the zero Poisson's ratio in Refs. [Science 235, 1038-1040 (1987), Science 320, 504-507 (2008)]. In conclusion, the mechanical properties of a monolayer are impacted by confinement effects at interfaces when deposited on a substrate or encapsulated. The degree to which the negative or biased-Poisson's ratio is affected depends on various factors, including the substrate's nature, the interface properties, and the degree of material compatibility between the two. On the other hand, the dependence of negative or biased-Poisson's ratio on confinement effects offers new knobs to tune the mechanical properties of such 2D functional materials.

Fig. R4 (a) Strain Z versus Strain Y for PdO₂ in PdO₂/graphene HBL and PdO₂. (b) Strain Z versus Strain X for GeO₂ in GeO₂/HfSe₂ HBL and GeO₂.

In our revised manuscript, we have added the description regarding zero Poisson's ratio observed in WO₃, AgO and CuO monolayers at near zero strain, as shown below.

“Excitingly, we find that WO₃ (110), AgO (101) and CuO (101) monolayers expand in the z -direction, regardless of whether they are subjected to stretching or compression in the x/y direction, as depicted in Fig. 3(c, d). That is, positive Poisson's ratio (PPR) appears during compression and NPR appears during stretching. Notably, at near zero strain, their Poisson's ratios become zero, which is

analogous to the zero Poisson's ratio reported in Refs^{44, 45}. Additionally, in contrast to the previous findings on Pd-decorated borophene⁴⁶, this extraordinary mechanical effect observed in the WO₃, AgO and CuO monolayers is intrinsic and arises from their unique puckered structures, which we term the biased-NPR effect. The relevant deformation mechanism is illustrated in Supplementary Figures 38 and 39." (P. 14)

Comment 1.5: 5) *The authors mention negative Poisson's ratio but I see positive Poisson's ratio in Fig. 3e-f.*

Reply: Thank you for your comment. We would like to clarify that the auxetic monolayers we predicted exhibit an NPR in the out-of-plane z -direction (as shown in Fig. 3a-d in the main text), while displaying a positive Poisson's ratio (PPR) in the in-plane xy -direction (as illustrated in Fig. 3e-f in the main text). In addition, for WO₃, AgO, and CuO, PPR and NPR appear during compression and tension stages, respectively, which we refer to as biased-Poisson's ratio effect (as demonstrated in Fig. 3c-d in the main text).

To avoid any confusion or misunderstanding, we have made some modifications to emphasize that NPR exists in the out-of-plane direction, while PPR exists in the in-plane direction. Specifically, we have added the phrase "**In-plane PPR**" in Figure 3(e-f). Furthermore, we have included the phrases "**out-of-plane**", "**in-plane**", and "**in the out-of-plane direction**" in the appropriate places in pages 12-14 of the revised manuscript.

Comment 1.6: 6) *Use different symbols to indicate the different properties in Fig. 1 instead of just circles.*

Reply: We thank the referee for this comment. We comply with the referee's suggestion and have revised Fig. 1(e) in the manuscript as shown in Fig. R5 below:

Fig. R5 E_b versus interplanar spacing (d_{hkl}). Materials classified as easily exfoliable, potentially exfoliable and high binding energy are demonstrated in different colors. The selected 2D oxide dielectrics (five-pointed star), auxetic monolayers (circle), 2D valleytronic oxide (triangle) and the monolayer with the lowest E_b (rectangle) are marked by different symbols and chemical formulae.

Reviewer #2 (Remarks to the Author):

Comment: *The authors discuss an approach to extract 2D materials out of parent ones where the parent material is not layered. They focus in particular on oxides with the aim of identifying high- k dielectrics for e.g. heterostructure applications in nano electronics. The approach is based on gradually expanding the lattice parameter along one direction and then computing the binding energy while relaxing atoms (and where two different qualitative behaviours are identified, depending on whether bonds reconstruct). The approach is different from papers in the literature that e.g. start from layered parent compounds (e.g. *Nature Nanotech* 13, 246–252 (2018)) or combinatorial lattice decoration (e.g. *J. Phys. Chem. C* 119, 13169–13183 (2015)). The identified layers are then investigated in detail identifying candidates with interesting mechanical and dielectric properties. While the paper provides some potentially interesting results, I have some questions for the authors both to better understand the novelty of the results, and to understand the realistic possibility of realising such materials, before I can recommend it for publication.*

Reply: We sincerely thank the referee for carefully reviewing our work and providing valuable suggestions to improve the manuscript.

Comment 2.1: - *Can the authors describe in more detail the exact algorithm used to extract the 2D materials? For instance:*

- *how many steps are considered for the c lattice parameter and with which*

step size

- is a relaxation is performed at every step? Is the relaxation for everything (in-plane cell+atoms), or only atoms, or with some additional constraints, ... In particular, if the in-place cell is relaxed or not can significantly affect the results.

- is this approach novel or it was already used in the literature?

- can the authors add a figure at least for the few most relevant materials, similar to Figs. 1a and 1d, but with actual numbers and units on the axes to have a more quantitative understanding of the approach?

Reply: We thank the referee for these valuable comments.

For all the exfoliation cases we considered, it typically takes 22 to 160 stretching steps with steps of 5% strain to complete the extraction of 2D material from its 3D precursor.

The relaxation is performed at each step, during which the in-plane lattice constants a , b and all the atomic positions in the system are fully optimized.

For the approach used in our study, we will first give a short description and more detailed explanation with a specific example will follow.

For the extraction of 2D materials from non-layered bulks, the first step is to determine the promising exfoliated crystal planes. The strategy is to identify the crystal planes with large difference between in-plane and interplanar interactions. A primary tool is the set of interplanar spacing (d_{hkl}) and packing ratio of different planes. We select the close-packed planes and nearly close-packed planes with large d_{hkl} , indicating that there may be a relatively weak out-of-plane interaction and strong in-plane bonding. In the second step, we rotate the selected crystal plane (hkl) to the (001) plane, in preparation for the subsequent exfoliation calculations. In the last step, we extract 2D material from its bulk precursor by gradually increasing the tensile stress on the selected crystal plane.

Secondly, we note that one important contribution of our work is proposing a new approach for the high-throughput extraction of 2D materials from nonlayered crystals. This involves screening non-layered crystallographic planes that have the potential to be exfoliated based on a geometric criterion, followed by conducting vdW-DFT calculations of the exfoliation process. The approach is independent of chemical composition, bonding type and symmetry, and hence can be extended to any nonlayered crystal, providing a flexible and cost-effective tool for mining 2D materials from largely untapped nonlayered bulk materials space, and offering the opportunity to study the potentially rich physics in 2D materials. We believe that our paper can be very interesting for those who are working on 2D-materials-related studies, and can surely stimulate further studies along this line.

A specific example: HfO₂

We take HfO₂ as an example to explain the exfoliation procedure in greater detail. The crystal structure of HfO₂ is shown in Fig. R6 (a).

For the purpose of identifying the promising exfoliated crystal planes, we calculate the close-packed degree and interplanar spacing (d_{hkl}) of 1113 crystal planes. As shown in Fig. R6 (b), the positive correlation between the close-packed degree and d_{hkl} is clearly noticeable. That is, a close-packed plane often has a large interplanar spacing. The crystal plane $(\bar{1}11)$ is selected as the optimal candidate for the exfoliation calculations since it satisfies both large d_{hkl} (3.17 Å) and close packing.

In addition, close-packed planes usually have the lowest surface energy, and hence an isolated crystal in equilibrium tends to be bounded by close-packed planes. Interestingly, the pure HfO₂ skin on the interfacial of liquid alloy was identified as $(\bar{1}11)$ crystal plane in the experiment (see Fig. R6 (c)) [Science **358**, 332-335 (2017)]. Our predicted close-packed crystal plane $(\bar{1}11)$ by geometric criteria matches perfectly with the experimental result, validating the approach.

[FIGURE REDACTED]

Fig. R6. (a) Crystal structure of HfO₂. (b) Our work: % close-packed versus d_{hkl} . (c) Ref. [Science **358**, 332-335 (2017)]: Characterization of HfO₂ nanosheet derived from the exfoliation method: TEM characterization (left), selected-area electron diffraction (right top) and HRTEM images (right bottom; scale bar, 0.5 nm).

The second step is to rotate the $(\bar{1}11)$ close-packed crystal plane to the (001) crystal plane. In the last step, we calculate the interlayer binding energy by increasing the lattice constant c step by step, during which the in-plane lattice constants and all the atoms in the system are fully relaxed. As the lattice constant c

gradually increases, the interlayer bonding gradually weakens until separation, and then the separated part is reconstructed, eventually forming a 2D structure that does not change with stretching, as shown in Fig. R7 (a). The relevant energy curve is shown in Fig. R7 (b).

Fig. R7. (a) Procedure for calculating the interlayer binding energy by increasing the lattice constant, c . (b) Schematic illustration of a binding energy curve. For clarity, the energy and lattice constant c at zero strain are set to zero.

To our best knowledge, we are not aware of the use of the above noted approach in the literature.

Finally, to fully address the referee's comments, we revised our manuscript and Supplementary Information in the following way (the blue fonts are newly added or revised contents):

- 1) We added the following contents in the revised manuscript to assist readers in locating the detailed explanation regarding exfoliation procedure.

“Please refer to Part I.I of the Supplementary Information and Supplementary Figure 2 for a detailed explanation of the exfoliation procedure.” (P.4)

- 2) We added the following contents in the Part I.I of the Supplementary Information to provide further details on the extraction process of 2D materials from non-layered bulk materials.

“For the extraction of 2D materials from non-layered bulks, the first step is to determine the promising exfoliated crystal planes. The strategy is to identify the crystal planes with large difference between in-plane and interplanar interactions. A primary tool is the set of interplanar spacing (d_{hkl}) and packing ratio of different planes. We select the close-packed planes and nearly close-packed planes with large d_{hkl} , indicating that there may be a relatively weak out-of-plane interaction and strong in-plane bonding. In the second step, we rotate the selected crystal plane (hkl) to the (001) plane, in preparation for the subsequent exfoliation calculations. In the last step, we extract 2D material from its bulk precursor by gradually increasing the tensile stress on the selected crystal plane. The specific procedure is as follows: we stretch the crystal along the z -direction in steps of 5% strain by increasing the

lattice constant c , during which the in-plane lattice constants a , b and all the atoms in the system are fully optimized. In addition, to complete the extraction of 2D material from its 3D precursor, it usually takes 22 to 120 stretching steps, for all the exfoliation cases we considered.”

- 3) We added the specific numbers and units on the axes of Fig. 1(a, d) in the revised manuscript. The relevant legend is also updated.
- 4) We added Supplementary Figure 2 (as shown below) to display the energy evolution and crystal structure changes during the extraction process of our most promising 2D dielectric oxides GeO_2 and Fe_2O_3 from their nonlayered bulks.

Fig. R8. Binding energy curve for (a) GeO_2 (101) and (b) Fe_2O_3 (101). For clarity, the energy and lattice constant c at zero strain are set to zero. Inset: The crystal structures correspond to the blue diamonds on the energy curve from left to right.

Comment 2.2: - *How confident can we be that the layers can actually be exfoliated? My doubt is the following. Even if the binding energy is low, do we know that, in plane, the binding energy is still high? If it is small also in plane, the layer would "break" when trying to exfoliate it. The authors make some arguments based on the packing ratio, that intuitively might make sense. However, looking at e.g. the bulk structure of GeO_2 and just looking at the bond lengths, it's not obvious that the in-plane bonds remain very strong. This is of course one of the arguments of the authors. Can they test this? E.g. trying to "break apart" some of the best candidates in-plane, to see if they are still strongly bounded, e.g. computing the binding energy also in plane once the 2D layer is formed? Or use some other approach to assess their actual stability (e.g. some short molecular dynamics at room temperature).*

Reply: We thank the referee for their insightful comment and valuable suggestion regarding the actual stability of monolayers.

Firstly, to determine whether the tightly bonded close-packed plane would maintain its strength *once the 2D layer is formed*, we examined the in-plane binding energy

(E_b) of our most promising 2D dielectric oxides, GeO_2 and Fe_2O_3 by stretching the crystals along the y -direction in steps of 5% strain. For comparison, we also calculate the in-plane E_b of graphene and h-BN. The results, depicted in Figure R8, indicate that the E_b value of GeO_2 ($32 \text{ J}\cdot 10^{-10}/\text{m}$) is markedly higher than that of h-BN ($10.8 \text{ J}\cdot 10^{-10}/\text{m}$) and graphene ($13.7 \text{ J}\cdot 10^{-10}/\text{m}$). Furthermore, Fe_2O_3 's in-plane E_b ($12.5 \text{ J}\cdot 10^{-10}/\text{m}$) is comparable to that of h-BN and graphene. These findings suggest that the predicted 2D oxides have robust in-plane bonding.

Fig. R9. Binding energy curve versus the changes in lattice constant b for (a) GeO_2 (101), (b) Fe_2O_3 (101), (c) h-BN, and (d) graphene. The crystals are stretched along the in-plane y -direction. For clarity, the energy and lattice constant b at zero strain are set to zero. Inset: The crystal structures correspond to the blue diamonds on the energy curve from left to right.

Secondly, to test whether the in-plane bonds remain very strong in the monolayers at a finite temperature, we performed *ab initio* molecular dynamics (AIMD) calculations at room temperature according to the referee's suggestion. First, we pick up representative 2D oxides from each space group, including our most promising 2D dielectric oxides, auxetic monolayers, as well as other 2D magnetic/nonmagnetic semiconductors/metals. The AIMD results, presented in Figure R10 (see Supplementary Figs. 6-8, 10-16, 20-24, and 26-30 for details), show that the total energy fluctuation of most of monolayers is rather small throughout the simulation, and the final structures do not shatter and are only distorted a little, demonstrating that they are thermally stable and can exist quite stably at room temperature. However, there are exceptions such as the Ag_2O (111) prototype and VO_2 (022), whose final structures are evidently distorted and cannot be optimized back to their initial structures through structure relaxation. As a result, we have removed them from the list of predicted stable 2D oxides.

Fig. R10. Total system energy fluctuation with simulation time for the selected 2D oxides.

In summary, the evaluation of in-plane binding energy and AIMD simulations at room temperature both demonstrate the benign stability of the predicted 2D oxides. To fully address this referee's comments, we revised our manuscript and Supplementary Information in the following way (the blue fonts are newly added or revised contents):

- 1) We discussed the AIMD results of the monolayers in the revised manuscript as follows,

“Subsequently, we thoroughly evaluate the actual stability of 2D oxides with stable phonon dispersion. We initiate this process by performing *ab initio* molecular dynamics (AIMD) calculations. The results shown in Supplementary Figs. 6-8, 10-16, 20-24, and 26-30 show that for most of monolayers, the total energy fluctuation throughout the simulation is rather small, and the final structures do not shatter and are only distorted a little, demonstrating that they are thermally stable. However, there are exceptions such as the Ag₂O (111) prototype and VO₂ (022), whose final structures are evidently distorted and cannot be relaxed back to their initial structures, which are not included in the list of predicted stable 2D oxides.” (P. 7-8)

“We plot the polar histogram in Fig. 1(f) to show the predicted stable EE and PE 2D oxides, comprising 14 space groups that include non-centrosymmetric and polar as well as chiral structures (see Supplementary Table 22).” (P.8)

2) We supplemented the calculation details of AIMD in section **Methods**:

“A supercell containing around 100 atoms for each 2D material with Gamma point K sampling in wave vector space is used for AIMD simulations. NVT ensemble simulations with a Nosé-Hoover thermostat at 300 K and a time step of 3 fs are carried out. The total simulation time is ~10 ps.” (P. 20)

3) We added Supplementary Figs. 6-8, 10-16, 20-24, and 26-30 (c), which offer AIMD results.

Comment 2.3: *Some additional comments:*

- when creating heterostructures, lattice-matching can be very important. Did the authors consider lattice-matching considerations in their analysis of heterostructures? If so, can this be better commented? If not, can the authors discuss the strain required to lattice-match the layers, and check the changes in properties (e.g. in the 2D materials alone) due to this strain?

Reply: We thank the referee for this comment. We have indeed considered lattice matching in the construction of the heterobilayer (HBL). We regret for not including these contents in the previous manuscript. In the revised manuscript, we have added the following contents regarding lattice matching,

“The relaxed lattice constants of GeO₂ and TMDs are provided in Supplementary Table 29. We use a 4×2 rectangular supercell of MoS₂ (MoSe₂) to match a 3×1 GeO₂ supercell, and a 2×3 rectangular supercell of HfS₂ (HfSe₂) to match a 3×1 GeO₂ supercell. The strains applied to the GeO₂ along the x-direction for GeO₂/MoS₂, GeO₂/MoSe₂, GeO₂/HfS₂, and GeO₂/HfSe₂ HBLs are 0.09%, 1.47%, 0.18%, and 0.23%, respectively, and the corresponding values along the y-direction are 2.37%, 3.87%, 2.12%, and 2.62%, respectively.” (P. 15)

We added Supplementary Table 29 (as presented in Table R4) to provide the relaxed lattice constants of GeO₂ and TMDs.

Table R4. The relaxed lattice parameters for GeO₂ and the rectangular cell of transition metal dichalcogenides.

TMDs	a (Å)	b (Å)	γ (°)
GeO ₂	4.01	10.9	90
MoS ₂	3.15 ¹³⁻¹⁵	5.46	90
MoSe ₂	3.24 ¹³⁻¹⁵	5.61	90
HfS ₂	6.27	3.62	90
HfSe ₂	6.33	3.65	90

To investigate the impact of lattice mismatch on GeO₂ properties, we calculated the band gaps and static dielectric constants of the GeO₂ under different strains in the

x -direction. Our findings, presented in Table R5, reveal that the band gap of GeO₂ is perfectly preserved in the strain range we consider, while the static dielectric constant of GeO₂ is highly sensitive to strain. However, the ultra-high k value of GeO₂ can be maintained under a small strain. It is worth noting that the static dielectric constant of GeO₂ increases significantly under a 0.5% strain. Furthermore, we also calculated the Poisson's ratio of GeO₂ on the HfSe₂ substrate. Our findings shown in Fig. R11 indicate that for GeO₂ in the HBL, the pristine monolayer's NPR effect transforms into a biased-Poisson's ratio effect, with PPR appearing during compression and NPR appearing during stretching. Remarkably, at near zero strain, the Poisson's ratio for GeO₂ in HBL becomes zero, analogous to the zero Poisson's ratio in Refs. [Science 235, 1038-1040 (1987), Science 320, 504-507 (2008)].

Table R5. The band gaps (E_g) and static dielectric constants (k) of GeO₂ under different strains along the x -direction.

Strain X (%)	E_g	k
0	3.29	99
0.5	3.41	151.2
1	3.42	29.9
1.5	3.43	19.4
2	3.42	14.8

Fig. R11. Strain Z versus strain X for GeO₂ in GeO₂/HfSe₂ HBL and pristine GeO₂ monolayer axis.

In the revised manuscript, we added the following contents to illustrate the impact of strain to the band gaps, static dielectric constant, and Poisson's ratio of 2D dielectric oxide GeO₂:

“To evaluate the impact of lattice mismatch on GeO₂ properties, we calculate the band gaps and static dielectric constants of the GeO₂ unit cell under different strains in the x -direction. Our calculated results, presented in Supplementary Table 32, reveal that the band gap of GeO₂ is perfectly preserved in the strain range we have considered, while the static dielectric constant of GeO₂ is highly sensitive to strain.

However, the ultra-high k value of GeO₂ can be maintained under a small strain. It is worth noting that the static dielectric constant of GeO₂ increases significantly under a 0.5% strain. Furthermore, we also compute the Poisson's ratio of GeO₂ on the HfSe₂ substrate. Our findings shown in Supplementary Fig. 44 indicate that for GeO₂ in the HBL, the pristine monolayer's NPR effect transforms into a biased-Poisson's ratio effect, with PPR appearing during compression, NPR appearing during stretching, and zero Poisson's ratio^{44,45} appearing at near zero strain." (P. 17-18)

Comment 2.4: - *The C2DB database (<https://cmrdb.fysik.dtu.dk/c2db>) provides some GeO₂ structures. Could the authors compare the structures they identify (not only GeO₂, also the other candidates) with those existing in the C2DB and assess if they are the same or they are different?*

Reply: Thanks for the nice suggestion. We followed the referee's suggestion and conducted a one-by-one comparison between our predicted stable 2D oxides and the structures in the C2DB database.

We will first give a brief summary. Our predicted stable 2D oxides correspond to 32 chemical formulas, out of which 12 can be found in the C2DB database, as illustrated in Table R6. It is worth noting that 8 of these 12 chemical formulas share the same lowest energy structure as our predicted oxides, confirming the validity of our exfoliation procedure. Additionally, it is important to highlight that the multifunctional 2D oxides, including high- k oxides (GeO₂, Fe₂O₃, RhO₂, WO₃), auxetic monolayers (GeO₂, RhO₂, WO₃), and 2D valleytronic oxide (BaO) that we have emphasized in the main text, are new structures that have not been reported in the C2DB database.

Next, we will provide a detailed explanation of the comparison process. First, we conduct a preliminary comparison of the lattice parameters and space group to determine whether the two crystals are identical. Then, we conduct a precise assessment of their identity by comparing the atomic positions of the two crystals. To accomplish this, we calculate the root-mean-square deviation (RMSD) of the atomic positions between the two structures. The RMSD is calculated as follows:

$$\text{RMSD} = \sqrt{\frac{1}{N} \sum_i^{n_{\text{atom}}} [(x_i - x_i')^2 + (y_i - y_i')^2 + (z_i - z_i')^2]} \quad (\text{R1})$$

where i cycles through all atoms, x_i and x_i' are the x-coordinates of the i^{th} atom in the two respective structures, respectively, and y and z are analogous. A low RMSD value indicates a high degree of similarity or identity between the two

structures. Utilizing this approach, we identified eight identical crystal structures with RMSD values ranging from 0.0006 to 0.0109. For further details, please refer to Supplementary Table 21.

Finally, we calculated the total system energy of our predicted oxides as well as those present in the C2DB database. The calculated results are depicted in Fig. R12, which clearly illustrates that most of our predicted oxides have lower energies compared to those in the C2DB database. In fact, nearly 83% of our predicted oxides, located on the left side of the red dashed line, have at least one structure with an energy that is either lower or equal to the energy of the lowest energy structure in the C2DB database. On the other hand, the remaining two oxides located on the right side of the red dashed line have slightly higher energies than the energy of the lowest energy structure in the C2DB database. Nonetheless, the stability of these two oxides has been confirmed by both phonon dispersions and *ab initio* molecular dynamics calculations.

Table R6. A summary for the comparison between 2D oxides we predict and those in the C2DB database.

Database	C2DB		
	Same (8)	Different (4)	Not have (20)
Our predicted oxides	AgO, CuO, HfO ₂ , PbO ₂ , PdO ₂ , PtO ₂ , SnO ₂ , ZrO ₂	Ga ₂ O ₃ , GeO ₂ , RhO ₂ , TiO ₂	BaO ₂ , BaO, BeO, CaO ₂ , CaO, CdO, Li ₂ O ₂ , MgO, SrO ₂ , SrO, WO ₃ , ZnO, CoO, Fe ₂ O ₃ , OsO ₂ , CsO ₂ , KO ₂ , NaO ₂ , PdO, RbO ₂

Fig. R12. The total system energy per atom comparison between 2D oxides (*h k l*) we predict and those in the C2DB database. For clarity, the energy of our predicted 2D oxide (*h k l*) with the lowest energy structure is set to be zero.

To fully address the referee's suggestion, we have revised our manuscript and Supplementary Information in the following way (the blue fonts are newly added or revised contents),

- 1) We added the following contents to the **Nonlayered Oxides Exfoliation** section in the revised manuscript:

“Encouraged by the aforementioned outcomes, we conduct a one-by-one comparison between our stable 2D oxides and the structures in the C2DB³⁴ database. This involves comparing the chemical formula, lattice parameters, space group, atomic position, and total system energy. The detailed comparison procedure is given in the Methods section. Our predicted stable 2D oxides correspond to 32 chemical formulas, out of which 12 can be found in the C2DB database. The comparison results, presented in Supplementary Table 21, indicate that out of the 12 chemical formulas, 10 have at least one structure with energy equal to or lower than the lowest energy structure in the C2DB database. Notably, there are 8 identical structures, which validates the reliability of our exfoliation procedure. Additionally, we emphasize that the multi-functional 2D oxides, such as auxetic high-*k* oxides (GeO₂, RhO₂, WO₃), AFM high-*k* oxide (Fe₂O₃), and 2D valleytronic oxide (BaO) that we discuss below, are new structures not previously reported in the C2DB database.” (P. 8-9)

- 2) We added a new subsection titled "**Comparison of Crystal Structures**" in the Methods section to illustrate the comparison method. (P. 20)
- 3) We added Table 21 in Supplementary Information to provide the comparison results between 2D oxides we predicted and those in the C2DB database.

Comment 2.5: - *Authors discuss the phonon dispersion of their selected materials. Are materials filtered based on whether the phonons are real and positive? As they discuss, most of them are, but a few have imaginary phonons ($\omega^2 < 0$), e.g. AgO (101), NiO (011), K2O (110), ... Are they discarded? If not, I suggest to use an approach such as the one of Togo, Tanaka, Phys. Rev. B 87, 184104 (2013): <https://journals.aps.org/prb/abstract/10.1103/PhysRevB.87.184104> - it is relatively easy to use and can provide stable structures (possibly in a larger supercell) starting from the unstable ones. Those with unstable phonons are unstable and therefore not the final structures (and this can affect e.g. their mechanical properties).*

Reply: We appreciate the valuable suggestion of the referee on searching the metastable structures associated with the unstable structures obtained from the exfoliation procedure. In the meantime, we would like to keep the scope of this study on the 2D structures that can be directly obtained from the exfoliation procedure. We thank the referee for bringing to our attention the fascinating work by Togo and Tanaka [PRB **87**, 184104 (2013)]. Their work suggests an algorithm for automated searching of the phase-transition pathway using first-principles calculations. The computational procedure consists of multiple calculation modules, each of which takes an initial crystal structure as input. The initial structure

undergoes complete relaxation before being converted to a supercell and used to compute force constants. If any instability is found, the crystal structure is distorted with collective displacements indicated by the instability, which will be used as an initial crystal structure of the next unit. The procedure is repeated until all the endpoint units show dynamically stable. By implementing this algorithm, Togo and Tanaka identify the dynamical stability and interrelationships of various structures generated from a simple cubic structure for seven metallic elements. In future work, we will use the approach described in Ref. [A. Togo and I. Tanaka, Physical Review B **87**, 184104 (2013)] to search for the metastable structures in large supercells.

In our revised manuscript, we have supplemented the following contents to describe unstable monolayers and provide readers with the method in Ref. [PRB **87**, 184104 (2013)] to find dynamic stable structures (the blue fonts are newly added or revised contents),

“The computed phonon dispersions, shown in Supplementary Figs. 5-30, reveal that most of the oxide monolayers presented non-imaginary frequency phonon dispersions, indicating dynamic stability, while the remaining about 21% of the monolayers exhibit negative phonon spectra, predicted to be unstable. For these unstable monolayers, one could search for their stable or metastable structures by investigating the phase-transition pathway²⁸. However, due to workload constraints, we exclude the unstable monolayers from further study in this work.” (P. 7)

Comment 2.6: - at lines 253-256, authors say "the product of m_{eff} and Φ_b is roughly proportional to E_g ": can they justify this? Also, then they replace $(m_{\text{eff}} \times \Phi_b)^{1/2}$ with E_g (without square root), so the two sentences are not consistent. Please explain in more detail.

Reply: We thank the referee for pointing this out. We regret for the inconsistency between the two sentences, specifically the first sentence is not precise due to the lack of clear expression. Specifically, we cited the methods from Ref. [NPG Asia. Mater. **7**, e190 (2015)], where $(m_{\text{eff}} \times \Phi_b)^{0.5}$ was replaced with E_g to simplify f_{FOM} as $E_g \cdot k$. In addition, it is generally observed that an increase in the dielectric constant k is accompanied by a decrease in the tunneling effective mass m_{eff} and injection barrier Φ_b [Microelectronic Engineering **72**, 257–262 (2004), IEEE T. Electron Dev. **50**, 1027-1035 (2003)]. The inverse relation between E_g and k is well-known. As a result, there exists a positive correlation between both m_{eff} and E_g , and Φ_b and E_g . In the revised manuscript, we have modified the relevant contents as follows,

“In principle, one can compute m_{eff} and Φ_b accurately. Here, considering of the computational burden and the fact that there exists a positive correlation between $(m_{\text{eff}} \times \Phi_b)^{0.5}$ and E_g , we simplify f_{FOM} as $E_g \cdot k^{36}$.” (P. 11)

Comment 2.7: - Lines 272-273: authors say that the Ge electronic states in the valence and conduction are mainly from Ge in the edge layer. However, From Fig. 2c, for the first peak in the valence bands, the two contributions are essentially the same (so 50% from edge and 50% from the middle layer). Please fix the discussion.

Reply: We thank the referee for the comment. Indeed, in the first peak in the valence bands shown in Fig. 2c, the contribution from Ge in the middle and edge layers is essentially equal, which is different from the other Ge electronic states in the valence and conduction bands. We have revised our manuscript accordingly (the blue fonts are newly added or revised contents):

“Moreover, the total density of states of Ge in the middle and edge layers shown in Fig. 2(c) reveals that Ge electronic states in the valence and conduction bands are mainly from Ge in the edge layer except for the first peak in the valence bands where the contributions from Ge in the middle and edge layers essentially equal.”
(P. 11)

Comment 2.8: - Line 275: authors speak of a "hybridisation between unusually large number of O *p* valence bands and...". How can we say that it's "unusually large"? Can they e.g. show what happens in the bulk and that a smaller number of states hybridise, or some other justification?

Reply: Thanks for the comment. Firstly, we would like to clarify that in our previous text, the phrase "unusually large" referred to the number of O *p* states in the valence bands relative to O *s* states, Ge *s* and *p* states of GeO₂ monolayer. We have now deleted the phrase "unusually large" in the revised manuscript to improve the accuracy of the statement.

We have followed the referee's suggestion and compared the DOS of 2D GeO₂ and bulk GeO₂, as shown in Fig. R13. It is evident that compared to bulk GeO₂, the band gap of 2D GeO₂ becomes narrower and the number of states near or at the VBM and CBM increases, both of which are expected to enhance the dielectric response of GeO₂ monolayer. Additionally, the projected DOS of GeO₂ monolayer shown in our main text exhibits strong cross-gap hybridization between the O *p* valence bands and Ge *s* and *p* conduction bands. This strong cross-gap hybridization leads to a highly enhanced Born effective charges, similar to those of sulfosalts, and is expected to result in a large static dielectric constant.

Fig. R13 Comparison of total DOS between GeO₂ monolayer and its bulk precursor.

To fully address the referee's comment, we revised our manuscript in the following way (the blue fonts are newly added or revised contents):

- 1) We added the inset in Fig. 2(d) in the revised manuscript, which offers the comparison diagram for the total density of states of GeO₂ monolayer and bulk GeO₂.
- 2) we added the following contents to describe the changes in the density of states of GeO₂ monolayer relative to bulk GeO₂:

“The band gap narrowing of the GeO₂ monolayer is accompanied by an obvious increase in the number of states near or at the valence/conduction bands maxima/minima compared to the bulk GeO₂, as shown in the inset of Fig. 2(d). The projected density of states (PDOS) of GeO₂ monolayer shown in Fig. 2(d) shows strong cross-gap hybridization between the O *p* valence band states and Ge *s* and *p* conduction band states.” (P. 11)

Comment 2.9: - Which 2D layers have a dipole along *z*? For those, do the authors use a correct screening of the long-range electrostatic field? This can be very important to remove spurious effects due to the PBC.

Reply: We thank the referee for bringing this to our attention. We sincerely apologize for the omission of the calculation details in the Methods section of our previous manuscript. We have indeed taken into account the screening of the long-range electrostatic field in our predicted polar 2D oxide WO₃ (1 1 0), which is the only one with a dipole along the *z*-axis. To ensure accuracy, we first optimized the crystal structure of WO₃ (1 1 0) using a dipole correction method by setting "LDIPOL, IDIPOL, and DIPOL" parameters. Our planar average potential calculations, presented in Fig. R14, demonstrate that the dipole correction effectively eliminate erroneous electrostatic coupling between periodic replicas in the *z*-direction. Furthermore, all physical properties calculations of WO₃ (1 1 0) employed the same parameter set.

Fig. R14. The planar average of the potential for WO_3 (1 1 0)

To fully address this valuable comment, we have supplemented the following contents in our revised manuscript and Supplementary Information (the blue fonts are newly added or revised contents),

- 1) We supplemented the following contents to the calculation details of **Methods** section:

“The dipole correction^{62, 63} is used to eliminate spurious electrostatic coupling between periodic copies in the z -direction for polar material.” (P. 20)

- 2) We have highlighted the polar non-centrosymmetric monolayer in bold font in Supplementary Table 20.

Comment 2.10: - Do the authors use a correct treatment of the dielectric screening for 2D materials? This can significantly change the relevant dielectric properties of the material. See e.g. Sohier et al, *Nano Lett.* 17, 3758 (2017), <https://pubs.acs.org/doi/pdf/10.1021/acs.nanolett.7b01090>; the effects can be very important, with a different treatment of screening and the disappearance (in 2D) of the well-known LO-TO phonon splitting (that exists instead in 3D). In particular: is ϵ_{2D} of Suppl. Figures 32, 33 defined in the same sense of ϵ_{2D} in the citation above (Sohier 2017, Eq. (3))?

Reply: We express our gratitude to the referee for raising the issue of dielectric screening in 2D materials and for bringing to our attention the remarkable study by Sohier et al. [*Nano Lett.* **17**, 3758 (2017)]. This study employed Coulomb cutoff within density-functional perturbation theory to tackle the erroneous interaction between repeated images, effectively reproducing the accurate behavior of optical phonons in 2D materials. We would like to mention that density functional perturbation theory in the VASP package deals with the dielectric screening of 2D materials by utilizing a finite-size supercell approach to simulate periodic boundary conditions. To gauge the influence of error propagation resulting from long-range Coulomb interactions on the outcomes of DFPT calculations in the VASP package, in our original text, we selected eighteen 2D dielectrics and performed static

dielectric constant calculations with vacuum sizes of 15 and 30 Å (see Supplementary Fig. 35). The dielectric constants obtained for both vacuum sizes exhibited satisfactory agreement, implying that the dielectric constants acquired for a vacuum size of 15 Å were not impacted by environmental screening from periodic images. Furthermore, Osanloo et al. utilized the supercell approach within the density-functional perturbation method in the VASP package to investigate the static dielectric constants of rare-earth oxyhalides [Nature Communication **12**, 5051 (2021)]. Through their computational tests, they determined that a vacuum thickness of 15 Å was adequate. Finally, it is worth noting that the use of the supercell approach for calculating static dielectric constants is a common practice in literature, as evidenced by publications such as npj 2D Materials and Applications **2**, 6 (2018), and ACS Appl. Electron. Mater. **5**, 623–631 (2023).

We note that ϵ_{2D} in the work by Sohler et al. [Nano Lett. **17**, 3758 (2017)], is defined as $\epsilon_{2D}(|q_p|) = \epsilon_{ext} + r_{eff}|q_p|$, which is the wavevector-dependent, dielectric screening. However, our static dielectric constant does not include the nonlocal, namely wavevector-dependent, dielectric screening. Nonlocal effect is important for excitons and other quasiparticles in 2D materials. However, it was demonstrated that nonlocal effect is not an issue for the static dielectric constants obtained by VASP [Nature Communication **12**, 5051 (2021), ACS Appl. Electron. Mater. **5**, 623–631 (2023)]. We regret for the confusion in our presentation of static dielectric constant in previous Supplementary Figures 32, 33. In the revised version, we have replaced ϵ by k to unify the expression of static dielectric constant.

Comment 2.11: - *Authors say that GeO₂ forms Type-II heterostructures with HfS₂. However, the valence bands are essentially aligned, so it's ambiguous if it's type-I or type-II?*

Reply: Thank you for your feedback. As depicted in Fig. R15, there is a small difference of approximately 0.02 eV between the valence band maximum of GeO₂ and HfS₂. Consequently, in our previous version, we suggested a type-II interface band alignment between the two materials.

In the revised manuscript, we have modified the statement (as shown below) to provide more precision regarding the GeO₂/HfS₂ heterostructures (the blue fonts are newly added or revised contents):

“It is evident that GeO₂ forms straddling gap or type-I band alignment with MoS₂, MoSe₂ and HfSe₂ monolayers. However, in the case of the GeO₂/HfS₂ HBL, the valence band maximum of GeO₂ is only slightly larger than that of HfS₂, by approximately 0.02 eV, which falls within the numerical error range. Therefore, it is not possible to determine whether it is a type-I or type-II alignment.” (P. 16)

Fig. R15. Band alignment of GeO₂ and HfS₂ monolayers before and after forming hetero-bilayers.

Comment 2.12: - Line 394: authors say that the gap reduction is mostly caused by the drop of the CBM in MoS₂. However, from the band structure of Fig. 4c (and comparing with the band structure of pristine MoS₂ in the Supplementary), a big change is given by the "new" valence band at Gamma that has a much higher energy. So I am not sure one can say that the change is mostly due to the drop of the conduction band.

Reply: We thank the referee for the comment. We have analyzed the electronic structure of MoS₂/GeO₂ hetero-bilayer very carefully. The MoS₂/GeO₂ hetero-bilayer can be represented by the "-+-00" charge-layers, where "-" represents the anionic S layer, "+" represents the cationic Mo layer, and "0" represents the charge neutral GeO₂ layer. Due to the shielding effect of the "-+-" charge-layers, the interface electric field only exists in GeO₂, as shown in Figure R16 (a). This interface electric field will pull down the energy bands of MoS₂ relative to those of GeO₂, and also move the VBM state of MoS₂ towards GeO₂: the VBM state of free-standing MoS₂ is mainly on Mo, and part of this VBM state is shifted to the S near GeO₂ in the hetero-bilayer, leading to a "new" valence band (see the highest valence band at Gamma point in Fig. 4(c) of the main text). As pointed out by the referee, the reduction of MoS₂ band gap in the hetero-bilayer is mainly due to the emergence of the "new" valence band.

Fig. R16. Average potential versus coordinate z for (a) $\text{GeO}_2/\text{MoS}_2$, (b) $\text{GeO}_2/\text{HfS}_2$, (c) $\text{GeO}_2/\text{MoSe}_2$, and (d) $\text{GeO}_2/\text{HfSe}_2$ hetero-bilayers.

Comment 2.13: - Lines 394-399: the discussion is very interesting. However I think it should be better clarified (it's not so clear from the text, one has to think about it) that atoms on the edge are those mostly affected by the heterostructuring w.r.t. atoms in the middle of the 2D layer, and therefore the effect in the materials discussed is different depending on the projection of near-gap states on the edge or middle atoms.

Reply: We thank the referee for providing positive feedback on our discussion in previous lines 394-399 (“very interesting”). After carefully considering the comments 2.12 and 2.13, we have gained a deeper understanding of the gap changes of TMD in hetero-bilayer. Specifically, the charge arrangement from TMD to GeO_2 is “-+00” layered, where “-” represents the anionic S (Se) layer, “+” represents the cationic Mo (Hf) layer, and “0” represents the charge neutral GeO_2 layer. Due to the shielding effect of the “-+” charge layer, the interface electric field only exists in GeO_2 , as shown in Figure R16. This interface electric field will pull down the energy bands of TMD relative to those of GeO_2 , and also move the VBM state of TMD towards GeO_2 , see Figure R17. The VBM state of free-standing MoS_2 (MoSe_2) is on Mo, and part of this VBM state is shifted to the S (Se) near GeO_2 in the hetero-bilayer, leading to a “new” valence band (see the highest valence band at Gamma point in Fig. 4(c, e) in the main text). Conversely, the VBM state of free-standing HfS_2 (HfSe_2) is on S (Se) and becomes more localized on the S (Se) near GeO_2 in the hetero-bilayer. This enhances the quantum constraint and results in a decrease in the hole energy level, leading to an increase in the band gap.

Fig. R17. PDOS Comparison diagrams of (a) MoS₂ in HBL and pristine MoS₂, (b) HfS₂ in HBL and pristine HfS₂, (c) MoSe₂ in HBL and pristine MoSe₂, and (d) HfSe₂ in HBL and pristine HfSe₂. E_{vac} denotes the vacuum energy level.

In the revised manuscript, we have modified the explanation (as shown below) regarding the gap changes of TMD in HBL (the blue fonts are newly added or revised contents),

“The charge arrangement from TMD to GeO₂ is “-+00” layered, where “-” denotes the anionic S (Se) layer, “+” denotes the cationic Mo (Hf) layer, and “0” denotes the charge neutral GeO₂ layer. Due to the shielding effect of the “-+” charge layer, the interface electric field only exists in GeO₂ (see Supplementary Figure 43). This field will pull down the energy bands of TMD relative to those of GeO₂, and also move the VBM state of TMD towards GeO₂. For pristine MoS₂ (MoSe₂), the VBM state resides on Mo. However, in the hetero-bilayer, part of this state shifts towards the S (Se) near GeO₂, leading to an emerging valence state at Gamma point (see the highest valence band at Gamma point in Figure 4(c, e)), reducing the band gap. Conversely, for pristine HfS₂ (HfSe₂), the VBM state is on S (Se) and becomes more localized on the S (Se) near GeO₂ in the hetero-bilayer. This localization enhances the quantum constraint and decreases the hole energy level, leading to an increase in the band gap.” (P. 17)

Comment 2.14: - line 426 (methods): authors used DFT+U. Which values of U did they use, for which atoms, and how did they choose this value(s)?

Reply: Thanks for the question. In our work, we use U values for d orbitals that are fully tested by fitting to experimental lattice constants and magnetic moments.

We regret for not providing detailed information on U values in the previous Supplementary Information. In the current supplementary Information, we have included a full list of the U values we used, as well as a comparison of our calculated lattice constants and magnetic moments with the experimental values. Please refer to Supplementary Table 1. Correspondingly, we added the following contents in the revised manuscript to explain our use of U values (the blue fonts are newly added or revised contents),

“The U values are energy corrections for the spurious self-interaction energy introduced by GGA. We use U values for *d* orbitals that are fully tested by fitting to experimental lattice constants and magnetic moments, as evidenced in Supplementary Tables 1, 2.” (P. 20)

Comment 2.15: - *In the supplementary (workflow section), they say that they added oxide dielectrics TiO₂, ZrO₂, HfO₂. Why? They were not in the original database? Why they had to be added by hand?*

Reply: Thank you for your question. We implemented a filtering procedure that automatically fetches materials with no more than 10 atoms in the unit cell from the input database in consideration of the computational cost. As a result, materials such as TiO₂, ZrO₂, and HfO₂ were excluded from our initial list.

We then added these oxides to our bulk list due to the following reasons. Firstly, 2D HfO₂ with ($\bar{1}11$) plane has been successfully synthesized in the experiment [Zavabeti et al. *Science* **358**, 332-335 (2017)], which perfectly corresponds to our predicted close-packed crystal plane by geometric criteria. Secondly, we note that significant attention has been paid to IV-B metal oxides, especially TiO₂, ZrO₂, HfO₂, for their potential as alternative gate dielectrics to replace SiO₂. Therefore, we have added them to our data list for further analysis and comparison purpose.

Comment 2.16: - *Can the authors comment on the structure of Ag₂O (111) (Suppl. Figure 29) - it seems that there are isolated atoms, or this is just what it looks like from the visualisation? What is the binding energy for removing those atoms? (i.e. are they really part of the 2D layers?)*

Reply: We thank the referee for this comment. The referee’s observation that there are isolated Ag atoms in Ag₂O (111) is correct, which exist independently without any bonding to the other portion of the 2D layer. We have removed Ag₂O (111) from the list of predicted stable 2D oxides in our revised manuscript.

Our calculations show that 1.35 J/m² of binding energy is required to remove isolated Ag atoms, as displayed in Figure R18. Specifically, as external exfoliation stress increases, the total system energy gradually rises. But when the stress reaches

a certain level, the total system energy drops sharply. This is due to the formation of a new Ag-O bond between the isolated Ag atom and the nearest neighboring O atom (as seen in the third inset in Figure R18). We then further increase the exfoliation stress until the extraction of Ag atoms is completed, i.e., the total system energy converges.

Fig. R18. Variation of the total system energy for removing the isolated Ag atom. For clarity, the energy and z coordinate at zero strain are set to zero. The inserted crystal structures correspond to the blue diamonds from left to right.

We have added the following contents to the legend of Supplementary Figure 28 to provide a description of the isolated Ag atoms.

“There are two positions of Ag in Ag₂O (111): one bonds to oxygen atoms to form a puckered hexagon, and the other exists in isolation at the center of the hexagon”

We have added the following contents to the revised manuscript to provide a description of the AIMD calculation results regarding Ag₂O (111).

“The results shown in Supplementary Figs. 6-8, 10-16, 20-24, and 26-30 show that for most of monolayers, the total energy fluctuation throughout the simulation is rather small, and the final structures do not shatter and are only distorted a little, demonstrating that they are thermally stable. However, there are exceptions such as the Ag₂O (111) prototype and VO₂ (022), whose final structures are evidently distorted and cannot be relaxed back to their initial structures, which are not included in the list of predicted stable 2D oxides.” (P. 7-8)

Comment 2.17: - *In addition, I'd like to recommend some improvements on the language to the authors. Here just a few examples of sentences that do not sound correct in English (there are quite a few more):*

- *Line 66: geometrically screening potentially exfoliable crystallographic plane -> this sentence is incorrect from a grammatical point of view and not very clear*
- *Line 95: criteria -> criterion*
- *Line 104: is independently -> is independent*
- *Line 105: to any nonlayered crystals -> to any nonlayered crystal*

- Line 300: *if a stretched/compressed [...] is applied -> if a stretch/compression [...] is applied*
- Line 303: *possess [...] higher shear resistant -> resistant is not the correct word here!*
- ...

Reply: We thank the referee for his/her meticulous review to improve our presentation and make our work more rigorous. We have revised our manuscript in the following way (the blue fonts are newly added or revised contents):

- 1) We have revised the sentence of “... *geometrically screening potentially exfoliable crystallographic plane...*” as follows,

“To address this challenge, we propose a novel route to design 2D oxides from nonlayered precursors. This involves geometric screening of promising exfoliated crystallographic planes from nonlayered oxides and conducting van der Waals (vdW) DFT calculations of the exfoliation process.” (P. 2)

- 2) We have corrected a number of misspellings and inappropriate descriptions in the revised manuscript, including those pointed out by the referee. We would like to mention that we have removed the article "a" from the phrase "a geometric criteria" in the previous lines 94-95 and have modified "is independently" to "are independent" in the previous line 104.
- 3) We have rephrased “higher shear resistant” to “**higher shear modulus and stronger indentation resistance**”, as follows,

“GeO₂ monolayers possess more superior toughness⁴¹, higher shear modulus⁴² and stronger indentation resistance⁴³ compared to black phosphorus.” (P. 13)

Comment 2.18: *Some final comments:*

- the authors use interchangeably k and ϵ . Do they represent the same quantity, and if so can the same symbol be used? or can the difference be clarified?

Reply: Thanks for the careful reading of our paper. We regret for the confusion in our presentation of static dielectric constant in previous Supplementary Figures 32, 33. In the current version, we have replaced ϵ by k to unify the expression of static dielectric constant (see current Supplementary Figures 33, 34).

Comment 2.19: *- some supplementary tables and figures are not cited (e.g. it would be useful to cite Suppl. Fig. 37 when discussing the anomalous auxetic effect of WO₃)*

Reply: We thank the referee for this comment. We regret for our inconspicuous

citations. We would like to mention that we cited Supplementary Figure 37 in the sentence “*We offer a mechanistic explanation to understand the above anomalous mechanical effects and summarize them in the Supplementary Figs. 35-38.*” at line 329 in our previous manuscript.

To fully address this comment, we rephrased or added the contents regarding the citation of Supplementary Information, as shown below,

“We start with a set of nonlayered binary metal oxides in Materials Project²⁵ (see workflow in Supplementary Information and Supplementary Figure 1 for filtering details) and then utilize geometric criteria to identify the promising exfoliated crystallographic planes.” (P. 3)

“Remarkably, GeO₂ monolayers manifest the interesting out-of-plane negative Poisson’s ratio (NPR) effect, which stems from the puckered structures (See Supplementary Figure 36 for details).” (P. 12)

“Our calculations show that the monolayers PdO₂, PbO₂, RhO₂ and OsO₂ display a more obvious auxetic effect in the out-of-plane direction, see Fig. 3(b). Their NPR values are approximately 6-16 times than that of black phosphorus. The deformation mechanism is given in detail in the Supplementary Figure 37.” (P. 13-14)

“Additionally, in contrast to the previous findings on Pd-decorated borophene⁴⁶, this extraordinary mechanical effect observed in the WO₃, AgO and CuO monolayers is intrinsic and arises from their unique puckered structures, which we term the biased-NPR effect. The relevant deformation mechanism is illustrated in Supplementary Figures 38 and 39.” (P. 14)

Comment 2.20: - *line 202 of the PDF: they mention MnO and 21 similar structures: shouldn't it be "19 similar structures"?*

Reply: We regret for the typo. In the revised manuscript, we have corrected “21 similar structures” to “18 similar structures”, where MnO has been removed from the list of predicted stable 2D oxides due to its potential instability under hydrogen environments.

Comment 2.21: - *in Fig. 3e and 3f, does the distance from the center of the colored bars represent the same value (Poisson ratio) as the color bar? Can the author clarify this better in the caption? In addition, if this is the case, I don't understand why at zero angle, in Fig. 3f, the curve for WO₃ is red (while for AgO is still yellow) but the one of WO₃ is inside (at smaller distance from the center).*

Reply: We thank the referee for bringing this issue to our attention. We regret for disregarding the distance mapping and only correlating the colors of rings with the value (Poisson's ratio) of the colored bars.

In the revised manuscript, we have updated Fig. 3(e, f), where the distance from the center to the colored rings corresponds to the value (Poisson's ratio) as indicated on the color bar.

Comment 2.22: - *supplementary table 2: add units for d_{hkl} , and add link to the section defining the % close-packed. Also, it spans a very large range, 8% to 100%. Can they show two extreme examples (8% and 100% for instance) and comment on this large range?*

Reply: Thanks for these comments. We regret for missing the unit for d_{hkl} in previous Supplementary Table 2. In the revised Supplementary Table 3 (previous Supplementary Table 2), we have added the unit "Å" for d_{hkl} , as well as added the content "The definition of % close-packed can be found from the Methods section in the manuscript" in the header to assist readers in locating the pertinent information in the main text.

In a given crystal, close-packed degree describes how close the crystal plane ($h k l$) is to the close-packed plane. Our protocol automatically constructs the close-packed plane (assumed plane) of a given crystal according to the smallest atomic spacing. For the BaO and CsO₂ nonlayered crystals, the smallest atomic spacings d_0 and d_1 are marked by black arrows in Figure R19. It is easy to see from Figure R19 that, in the BaO crystal, (110) crystal plane is exactly the assumed close-packed plane that our protocol constructs, therefore the close-packed degree is equals to 100%. However, in the CsO₂ crystal, the (1 1 $\bar{1}$) crystal plane is far from the assumed close-packed plane, and the close-packed degree is only 8.9%. In short, close-packed degree provides a reliable estimation of how close-packed a plane is in a given crystal rather than different crystals. Lastly, we kindly mention that the interplanar spacings d_{hkl} are comparable among different crystals.

Fig. R19. Crystal structures for (a) BaO and (b) CsO₂, where the smallest atomic spacings d_0 and d_1 are marked by black arrows, respectively. The close-packed planes BaO (110) and CsO₂ (11 $\bar{1}$) are marked by the orange panels.

Comment 2.23: - *I am not sure if this is just an issue of the conversion to PDF: but can the authors provide higher-resolution images? e.g. the spin arrows in Suppl. Fig. 39 (panels b and c) are not really clear with the resolution of the PDF file I have.*

Reply: Thanks for the comment. We have provided the higher-resolution images in the revised manuscript.

Reviewer #3 (Remarks to the Author):

Comment: *The manuscript presents a strategic discovery and design of oxide monolayers by exfoliating nonlayered oxides with potentially low exfoliation energy, resulting in the identification of 61 potentially exfoliable and dynamically stable 2D oxides. The predicted 2D oxide families are expected to possess promising dielectric, mechanical, magnetic, and optoelectronic properties for technical applications, which could greatly diversify existing 2D materials in terms of material classes and functionalities if they are successfully synthesized experimentally. Overall, given the potential impact of this work in the field of functional 2D materials, I recommend the publication of the manuscript, provided that the following comments are addressed properly.*

Reply: We sincerely thank the referee for the evaluation of our work and for the recommendation of publication of our work on Nature Communications. We have addressed the referee's comments appropriately and made the corresponding changes in the revised manuscript.

Comment 3.1: *1. One concern regarding the predictions of oxide monolayers from nonlayered oxides is the potential lack of surface inertness that could compromise their stability in practical applications. The authors should address this concern by providing examples demonstrating that the surfaces of these monolayers are stable enough against molecule absorption. Specifically, it would be helpful to know the typical hydrogen binding energy on these monolayers to assess their stability.*

Reply: We thank the referee for bringing up the issue of surface inertness. Indeed, the surface stability of 2D materials is critical for their subsequent device applications, which claims robust inertness against ambient molecules for the surface atoms. We have fully taken into account the referee's suggestions and conduct a thorough assessment of the stability of our predicted 2D oxides.

We start by calculating the binding energy of 2D oxides with adsorbed hydrogen molecule ($E_{\text{bind}}[\text{H}_2]$). First, we pick up representative 2D structures from each space group, including our most promising 2D functional oxides, as well as other 2D

magnetic/nonmagnetic semiconductors/metals. The calculation process begins by randomly distributing four hydrogen molecules onto the surface of the supercell, with an adsorption density of $\sim 10^{14}$ molecules per square centimeter and an average distance between the molecules of around 10 Å. The in-plane lattice constants a , b and all the atomic positions in the system are fully optimized until the final force on each atom is < 0.01 eV/Å. The calculation equation for E_{bind} per H_2 molecule is as follows [Physical Review B **100**, 085416 (2019), Angewandte Chemie **128**, 977-980 (2016)],

$$E_{\text{bind}} = \frac{E_{\text{tot}}^{\text{adsorbed 2D oxide}} - E_{\text{tot}}^{\text{pristine 2D oxide}} - N \times E_{\text{tot}}^{\text{isolated molecule}}}{N} \quad (\text{R2})$$

where the numerator terms on the right-hand side represent the total energies of 2D oxides with N adsorbed H_2 molecules, pristine 2D oxides, and N isolated H_2 molecules, respectively. Our calculated results presented in Fig. R20 indicate that the $E_{\text{bind}} [\text{H}_2]$ of our predicted 2D oxides, especially our highly promising 2D functional oxides such as GeO_2 , Fe_2O_3 , WO_3 , RhO_2 , OsO_2 , etc., ranges from -0.038 to -0.135 eV/ H_2 , which is moderate and comparable to that of common 2D materials like monolayer TMDs (-0.063 eV/ H_2) and monolayer BN (-0.067 eV/ H_2). For the materials with adsorption energies outside this range (SrO_2 , CaO_2 , VO_2 , and MnO), we have removed them from the list of predicted stable 2D oxides.

Secondly, to assess the stability of our predicted 2D oxides at finite temperature, we carry out *ab initio* molecular dynamics (AIMD) simulations for ~ 10 ps at 300 K. The calculated results, shown in Supplementary Figs. 6-8, 10-16, 20-24, and 26-30, indicate that the majority of monolayers exhibit minimal fluctuations in total energy throughout the simulation, and their final structures undergo only slight distortion instead of breaking apart, suggesting that they are thermally stable, with the exceptions of Ag_2O (111) prototype and VO_2 (022), which are not thermally stable and have been removed from the current list of predicted stable 2D oxides.

Finally, to further evaluate the resistance of our most promising 2D dielectric oxides (GeO_2 , Fe_2O_3 , WO_3 , RhO_2 , and SnO_2) to ambient molecules adsorption at finite temperature, we perform AIMD simulations at room temperature under various environments containing O_2 , H_2 , or H_2O . The results, depicted in Fig. R21, indicate that the surfaces remain stable without experiencing structural collapse due to molecule invasion or bond reconstruction, further confirming their ability to exist stably under ambient conditions.

In summary, the evaluation of hydrogen binding energy on the monolayers and AIMD simulations at room temperature both demonstrate the benign stability of our predicted 2D oxides against ambient molecule adsorption.

Fig. R20. Comparison of the typical hydrogen binding energy E_{bind} of the our predicted 2D oxides and monolayer TMDs.

Fig. R21. AIMD simulation at room temperature for the selected high- k 2D dielectric oxides with ambient molecule adsorptions, indicating stable surface without structural collapse.

To fully address this important comment, we have revised our manuscript and Supplementary Information in the following way (the blue fonts are newly added or revised contents),

- 4) We discussed the AIMD results and hydrogen binding energy on the monolayers in the revised manuscript as follows,

“Subsequently, we thoroughly evaluate the actual stability of 2D oxides with stable phonon dispersion. We initiate this process by performing *ab initio* molecular dynamics (AIMD) calculations. The results shown in Supplementary Figs. 6-8, 10-16, 20-24, and 26-30 show that for most of monolayers, the total energy fluctuation throughout the simulation is rather small, and the final structures do not shatter and are only distorted a little, demonstrating that they are thermally stable. However, there are exceptions such as the Ag₂O (111) prototype and VO₂ (022), whose final structures are evidently distorted and cannot be relaxed back to their initial structures, which are not included in the list of predicted stable 2D oxides. Additionally, we calculate the hydrogen binding energy ($E_{\text{bind}} [\text{H}_2]$) on the monolayers to assess their surface inertness. The results presented in Supplementary Figure 31 indicate that the $E_{\text{bind}} [\text{H}_2]$ of our predicted 2D oxides, especially our highly promising 2D functional oxides such as GeO₂, Fe₂O₃, WO₃, RhO₂, OsO₂, etc., ranges from -0.038 to -0.135 eV/H₂, which is moderate and comparable to that of common 2D materials TMDs (-0.063 eV/H₂). Monolayers with $E_{\text{bind}} [\text{H}_2]$ outside this range, namely SrO₂, CaO₂, VO₂, and MnO, are not included in the list of predicted stable 2D oxides. Finally, for our most promising 2D high-*k* dielectric oxides, namely GeO₂, Fe₂O₃, WO₃, RhO₂, and SnO₂, we perform AIMD simulations at room temperature under various environments containing water, oxygen, or hydrogen. The results, shown in Supplementary Figure 32, further confirm their ability to exist stably under ambient conditions.” (P. 7-8)

“We plot the polar histogram in Fig. 1(f) to show the predicted stable EE and PE 2D oxides, comprising 14 space groups that include non-centrosymmetric and polar as well as chiral structures (see Supplementary Table 22).” (P.8)

- 5) We supplemented the calculation details of $E_{\text{bind}} [\text{H}_2]$ and AIMD in section **Methods**:

“For $E_{\text{bind}} [\text{H}_2]$ calculations, we place four hydrogen molecules on the surface of the supercell in a random manner. The adsorption density is $\sim 10^{14}$ molecules per square centimeter and the average distance between the molecules is ~ 10 Å. The calculation equation for $E_{\text{bind}} [\text{H}_2]$ is as follows⁶⁷⁻⁶⁸,

$$E_{\text{bind}} = \frac{E_{\text{tot}}^{\text{adsorbed 2D oxide}} - E_{\text{tot}}^{\text{pristine 2D oxide}} - N \times E_{\text{tot}}^{\text{isolated molecule}}}{N} \quad (1)$$

where the numerator terms on the right-hand side represent the total energies of 2D oxides with *N* adsorbed H₂ molecules, pristine 2D oxides, and *N* isolated H₂ molecules, respectively.” (P. 20-21)

“A supercell containing around 100 atoms for each 2D material with Gamma point *k* sampling in wave vector space is used for AIMD simulations. NVT ensemble simulations with a Nosé-Hoover thermostat at 300 K and a time step of 3 fs are

carried out. The total simulation time is ~10 ps.” (P. 20)

- 6) We added Supplementary Figures. 6-8, 10-16, 20-24, and 26-30 (c), which offer AIMD results, along with Figure 32, showcasing AIMD results with molecule adsorption.
- 7) We added Supplementary Figure 31, which provides the results regarding hydrogen binding energy on the monolayers.

Comment 3.2: 2. *Many of the newly predicted oxide monolayers exhibit out-of-plane negative Poisson's ratio but positive in-plane Poisson's ratio. It would be helpful for the authors to comment on the difference between the oxide monolayers and those 2D materials that have been predicted to have in-plane negative Poisson's ratio, such as the ones reported by J. Pan et al. in npj Computational Materials 6, 154 (2020) and L. Yu et al. in Nature Communications 8, 15224 (2017). This comparison would provide valuable context and help readers better understand the significance of the current findings.*

Reply: We thank the referee for bringing to our attention the highly relevant fascinating research conducted by J. Pan et al. and L. Yu et al. [Nature Communications **8**, 15224 (2017) and npj Computational Materials **6**, 154 (2020)]. These studies highlight a group of auxetic two-dimensional materials, specifically transition metal dichalcogenides/selenides/halides, that exhibit a noteworthy in-plane negative Poisson's ratio (NPR), which arises from a combination of geometric and electronic structure effects. This is an exceptionally intriguing finding.

Compared to the in-plane NPR discussed in the literatures above, our out-of-plane NPR effect in metal oxide monolayers is independent of chemical elements. Therefore, we attribute our out-of-plane NPR effect to purely geometric factors. Additionally, the auxetic behaviors in our predicted transition metal oxide monolayers WO₃, AgO, and CuO are compelling. These materials exhibit positive and negative Poisson's ratio during compression and tension stages, respectively, which we refer to as the biased-Poisson's ratio effect (as demonstrated in Fig. 3c-d in the main text). Lastly, GeO₂ ($k \sim 99$), RhO₂ ($k \sim 62$), and WO₃ ($k \sim 30$) stand out with ultra-high k values, which is significantly higher than that of the highly regarded 2D dielectrics CaF₂ ($k \sim 6$) and β -Bi₂SeO₅ ($k \sim 22$) [Nature Electronics **5**, 643-649 (2022)]. The unusual auxetic behavior in combination with the remarkable dielectric properties of two-dimensional metal oxides (GeO₂, RhO₂, and WO₂) could lead to novel multi-functionalities.

To fully address this comment, we have added the following contents in our revised manuscript (the blue fonts are newly added or revised contents):

“There are two primary deformation mechanisms for 2D materials that exhibit auxetic effects. One is the result of pure geometric effects, such as the out-of-plane negative Poisson’s ratio (NPR) observed in black phosphorus¹². The other is attributed to the interplay of both geometric and electronic structure effects, observed in transition metal selenides, transition metal halides⁴⁷, and 1T-type transition-metal dichalcogenides⁴⁸. We note that the auxetic behaviors in our predicted oxide monolayers are independent of chemical elements. Therefore, we attribute the occurrence of out-of-plane NPR and biased-NPR effects to purely geometric factors. For further explanations, please refer to Supplementary Part III.”
(P. 14)

Comment 3.3: *3. Another question related to the manuscript is the effectiveness of the 2D structure discovery strategy employed. While the use of close-packed planes in nonlayered structures is an intelligent way to reduce the computational cost of the structure discovery process, it is possible that the final monolayer structures identified may already exist in many layered materials. In this case, there may be significant overlap between the final structures obtained from either layered or nonlayered initial structures. The authors should address this concern by checking the fraction of identified oxide monolayer structures that have already been included in other 2D material databases such as the Computational 2D Materials Database (C2DB), which is constructed based on layered bulk materials and elemental substitution. Such an analysis would help demonstrate the novelty of the discovered oxide monolayers and provide insights into the effectiveness of the discovery strategy employed.*

Reply: We would like to express our gratitude to the referee for acknowledging our approach as "intelligent" and providing valuable feedback. We have taken the referee’s suggestion into consideration and performed a meticulous comparison between our predicted stable 2D oxides and the structures present in the C2DB database.

We will first give a brief summary. Our predicted stable 2D oxides correspond to 32 chemical formulas, out of which 12 can be found in the C2DB database, as illustrated in Table R7. It is worth noting that 8 of these 12 chemical formulas share the same lowest energy structure as our predicted oxides, confirming the validity of our exfoliation procedure. Additionally, it is important to highlight that the multi-functional 2D oxides, including high-*k* oxides (GeO₂, Fe₂O₃, RhO₂, WO₃), auxetic monolayers (GeO₂, RhO₂, WO₃), and 2D valleytronic oxide (BaO) that we have emphasized in the main text, are new structures that have not been reported in the C2DB database.

Next, we will provide a detailed explanation of the comparison process. First, we conduct a preliminary comparison of the lattice parameters and space group to determine whether the two crystals are identical. Then, we conduct a precise

assessment of their identity by comparing the atomic positions of the two crystals. To accomplish this, we calculate the root-mean-square deviation (RMSD) of the atomic positions between the two structures. The RMSD is calculated as follows:

$$\text{RMSD} = \sqrt{\frac{1}{N} \sum_i^{n_{\text{atom}}} [(x_i - x_i')^2 + (y_i - y_i')^2 + (z_i - z_i')^2]} \quad (\text{R3})$$

where i cycles through all atoms, x_i and x_i' are the x-coordinates of the i^{th} atom in the two respective structure, respectively, and y and z are analogous. A low RMSD value indicates a high degree of similarity or identity between the two structures. Utilizing this approach, we identified eight identical crystal structures with RMSD values ranging from 0.0006 to 0.0109. For further details, please refer to Supplementary Table 21.

Finally, we calculated the total system energy of our predicted oxides as well as those present in the C2DB database. The calculated results are depicted in Fig. R22, which clearly illustrates that most of our predicted oxides have lower energies compared to those in the C2DB database. In fact, nearly 83% of our predicted oxides, located on the left side of the red dashed line, have at least one structure with an energy that is either lower or equal to the energy of the lowest energy structure in the C2DB database. On the other hand, the remaining two oxides located on the right side of the red dashed line have slightly higher energies than the energy of the lowest energy structure in the C2DB database. Nonetheless, the stability of these two oxides has been confirmed by both phonon dispersions and *ab initio* molecular dynamics calculations.

Table R7. A summary for the comparison between 2D oxides we predict and those in the C2DB database.

Database	C2DB		
	Same (8)	Different (4)	No have (20)
Our predicted oxides	AgO, CuO, HfO ₂ , PbO ₂ , PdO ₂ , PtO ₂ , SnO ₂ , ZrO ₂	Ga ₂ O ₃ , GeO ₂ , RhO ₂ , TiO ₂	BaO ₂ , BaO, BeO, CaO ₂ , CaO, CdO, Li ₂ O ₂ , MgO, SrO ₂ , SrO, WO ₃ , ZnO, CoO, Fe ₂ O ₃ , OsO ₂ , CsO ₂ , KO ₂ , NaO ₂ , PdO, RbO ₂

Fig. R22. The total system energy per atom comparison between 2D oxides (hkl) we predict and those in the C2DB database. For clarity, the energy of our predicted 2D oxide (hkl) with the lowest energy structure is set to be zero.

By the way, we would like to mention that the bulk oxides used in our study were obtained from the MP database. Upon checking the newly updated MP database, we found that mp-223 GeO₂ no longer satisfies the filtering condition of the lowest energy structure. Consequently, we have removed mp-223 GeO₂ from our bulk list, and only retained the relevant contents of mp-733 GeO₂, which is experimentally the most stable structure, in the revised version.

To fully address the referee's suggestion, we have revised our manuscript and Supplementary Information as follows (the blue fonts are newly added or revised contents),

- 1) We added the following contents to the **Nonlayered Oxides Exfoliation** section in the revised manuscript:

“Encouraged by the aforementioned outcomes, we conduct a one-by-one comparison between our stable 2D oxides and the structures in the C2DB³⁴ database. This involves comparing the chemical formula, lattice parameters, space group, atomic position, and total system energy. The detailed comparison procedure is given in the Methods section. Our predicted stable 2D oxides correspond to 32 chemical formulas, out of which 12 can be found in the C2DB database. The comparison results, presented in Supplementary Table 21, indicate that out of the 12 chemical formulas, 10 have at least one structure with energy equal to or lower than the lowest energy structure in the C2DB database. Notably, there are 8 identical structures, which validates the reliability of our exfoliation procedure. Additionally, we emphasize that the multi-functional 2D oxides, such as auxetic high- k oxides (GeO₂, RhO₂, WO₃), AFM high- k oxide (Fe₂O₃), and 2D valleytronic oxide (BaO) that we discuss below, are new structures not previously reported in the C2DB database.” (P. 8-9)

- 2) We added a new subsection titled "**Comparison of Crystal Structures**" in the

Methods section to illustrate the comparison method. (P. 20)

- 3) We added Table 21 in Supplementary Information to provide the comparison results between 2D oxides we predict and those in the C2DB database.

Comment 3.4: *4. The authors stated that only a few of the predicted oxide monolayers have been experimentally studied. Therefore, it would be helpful if the authors could provide some comments and guidance on how these oxide monolayers can be synthesized through viable routes. Such information would be useful for researchers interested in synthesizing these materials and could help accelerate the experimental validation of the predicted properties.*

Reply: We would like to express our sincere appreciation to the referee for the nice and insightful suggestion. In the revised manuscript, we have provided two synthesized methods that have the potential to be utilized for synthesizing all of the oxide monolayers that we predicted. The first method is the liquid metal-assisted exfoliation technique proposed by A. Zavabeti et al. in 2017 [Science **358**, 332-335 (2017)]. The second technique involves the formation of 2D transition metal oxide nanosheets with nanoparticles as intermediates, which was proposed by Yang Juan et al. in 2019 [Nature Materials **18**, 970-976 (2019)].

Firstly, to aid in understanding the liquid metal-assisted exfoliation technique, we will first provide an example of its successful application in the synthesis of HfO₂ ($\bar{1}11$). Subsequently, we will discuss how this technique can be extended to obtain other oxides that we predicted.

Specifically, Zavabeti et al. used liquid gallium-based alloy as a reaction solvent and co-alloyed elemental hafnium into the melt. A self-limiting interfacial oxide that consists exclusively of HfO₂ naturally forms at the metal surface under an oxygen-containing environment, since HfO₂ results in the greatest reduction of Gibbs free energy. This leads to a more stable state of gallium-based alloys. The interfacial HfO₂ shows a crystallographic plane of ($\bar{1}11$), which perfectly corresponds to our predicted close-packed crystal plane. This outcome was expected since close-packed planes typically have the lowest surface energy, and therefore an isolated 2D crystal in equilibrium tends to be bounded by close-packed planes. Finally, Zavabeti et al. successfully exfoliated 2D HfO₂ from the alloy droplet, as the liquid nature of the parent metal results in the absence of macroscopic forces between the metal and its oxide skin.

The aforementioned findings suggest that oxide layers on liquid metals can be manipulated by appropriately matching alloying elements and alloy solvents, based on the Gibbs free energy for oxide formation. We note that many of our predicted metal oxides have lower Gibbs free energy than that of Bi₂O₃ and SnO₂, as seen in

Figure R23. Thus, if using the liquid bismuth-tin alloy (138°C) as the solvent, many our predicted oxide close-packed planes can be accessed as 2D layers. These include GeO₂ (auxetic, high-*k*), Fe₂O₃ (antiferromagnetic, high-*k*), ZrO₂ and TiO₂ (high-*k*), BaO (valleytronic), and WO₃ (anomalous auxetic), among others. This expectation is based on the fact that the oxide that yields the greatest reduction in Gibbs free energy and the close-packed crystal plane with the lowest surface energy determine the interfacial oxide skin of metal.

Fig. R23. Gibbs free energy of formation for metal oxides. Oxides to the right of the red (blue) dashed line are expected to dominate the interface if liquid gallium (bismuth–tin) alloys are the solvent.

Secondly, another possible approach is to use 3D nanoparticles as intermediates to synthesize our predicted 2D metal oxides close-packed planes. Yang Juan et al. have exhibited that 3D nanoparticles are first formed from the molecular precursor solution and then transform into 2D nanosheets [Nature Materials **18**, 970-976 (2019)]. The thermodynamic driving force has been recognized as the mechanism for the 3D-to-2D transition. The close-packed plane with the lowest surface energy is expected to be dominant for the surface.

Finally, to fully address this very important comment, we revised our manuscript and Supplementary Information as follows (the blue fonts are newly added or revised contents):

- 1) We added the following contents to the newly added **Discussions** section in the revised manuscript:

“Recently, Zavabeti et al. have demonstrated that the alloying of elemental hafnium into a liquid gallium-based alloy results in the natural formation of a 2D interfacial oxide skin of HfO₂ ($\bar{1}11$) crystal plane on the metal surface when exposed to an oxygen-containing environment¹⁹. This is because the oxide that yields the greatest reduction in Gibbs free energy and the close-packed crystal plane with the lowest surface energy dominate the surface. Thus, when using the liquid bismuth-tin alloy

(138°C) as the solvent, many our predicted metal oxides close-packed planes, such as GeO₂, Fe₂O₃, BaO, CaO, BeO, MgO, SnO₂ and WO₃, etc., can be accessed as 2D layers. Supplementary Table 33 presents the Gibbs free energy of formation for our predicted metal oxide monolayers, which is lower than that of Bi₂O₃ and SnO₂. The remaining predicted 2D oxides with higher Gibbs free energy than Bi₂O₃ and SnO₂, such as RhO₂, AgO, CuO, PbO₂, ZnO, CoO, and NiO, etc., can be synthesized using nanoparticles as intermediates. Yang Juan et al. have exhibited that 3D nanoparticles are initially formed from the molecular precursor solution and then transform into 2D nanosheets due to the thermodynamic driving force⁵⁶. The close-packed plane with the lowest surface energy is expected to be dominant for the surface. We hope that our findings can stimulate additional experimental research into the 2D oxides we have identified, particularly in relation to GeO₂, Fe₂O₃, WO₃, and RhO₂.” (P. 18-19)

- 2) We added Table 33 in the Supplementary Information to provide Gibbs free energy of formation for metal oxides.

Comment: *I hope these comments will be helpful for the authors in improving the manuscript.*

Reply: We express our heartfelt gratitude to the third referee for the invaluable comments and feedback provided. His/Her contribution has significantly enhanced the quality of our paper, and we are deeply appreciative of his/her efforts.

REVIEWERS' COMMENTS

Reviewer #1 (Remarks to the Author):

The authors have provided a thorough response to our comments. We recommend publication provided that the authors address the following 2 comments:

- 1) The authors should further improve their description of the methodology. For example, no mention is made of spin-orbit coupling or whether the collinear approximation is used when calculating the magnetic moments.
- 2) My previous question about the long-range magnetic order related to the question if the order was in-plane or out-of-plane. If the authors perform collinear calculations, perhaps they can not determine which it is. In-plane magnetic order will not be long range. The authors should comment in the paper.
- 3) The authors should estimate transition temperature using more recent approaches compared to (or in addition to) the Ref. 38 they use in the paper. One way to estimate can be found in Eq. (3) of

<https://journals.aps.org/prresearch/abstract/10.1103/PhysRevResearch.3.043024>

Reviewer #2 (Remarks to the Author):

The authors have satisfactorily replied to all my questions, as well as performed several new calculations to address my questions. Therefore, I think that the paper can now be accepted for publication.

Reviewer #3 (Remarks to the Author):

The authors have provided a set of comprehensive studies and analyses based on the referee's comments. All the concerns have been addressed and the conclusions are supported by additional data. I would therefore recommend the publication of this interesting material discovery work in Nature Communications.

Comments and Author reply

Reviewer #1 (Remarks to the Author):

Comment: *The authors have provided a thorough response to our comments. We recommend publication provided that the authors address the following 2 comments:*

Reply: We sincerely thank the referee for the positive evaluation of our work. We have addressed the referee's comments appropriately and made the corresponding changes in the revised manuscript.

Comment 1.1: *1) The authors should further improve their description of the methodology. For example, no mention is made of spin-orbit coupling or whether the collinear approximation is used when calculating the magnetic moments.*

Reply: We are grateful to the referee for the valuable comment. We regret for not specifying whether spin-orbit coupling or collinear approximation was employed in our calculations. In the previous calculations, including the determination of magnetic moments, we used collinear approximation without considering spin-orbit coupling. However, in the current version, we have incorporated non-collinear DFT calculations specifically for 2D Fe₂O₃. To fully address this important comment, we added the following contents to the **Methods** section of the revised manuscript (the blue fonts indicate newly added or revised contents):

“All calculations are performed within density functional theory (DFT)⁶⁰ in the Vienna Ab-initio Simulation Package (VASP)⁶¹ using the projector-augmented wave (PAW) pseudopotentials⁶² with the generalized gradient approximation GGA/PBE+U⁶³ exchange-correlation functional without considering the spin-orbit coupling effect.” (P. 20)

“We employ the collinear approximation for the relevant DFT calculations of magnetic materials, except for the calculation of magnetic anisotropy in Fe₂O₃.” (P. 20)

Comment 1.2: *2) My previous question about the long-range magnetic order related to the question if the order was in-plane or out-of-plane. If the authors perform collinear calculations, perhaps they can not determine which it is. In-plane magnetic order will not be long range. The authors should comment in the paper.*

Reply: We would like to express our gratitude to the referee for providing further feedback regarding the previous question concerning the long-range magnetic order of Fe₂O₃. Based on our latest non-collinear DFT calculations, it has been determined that the 2D Fe₂O₃ exhibits an out-of-plane easy axis and a magnetic

anisotropy energy (MAE) of 50 $\mu\text{eV}/\text{Fe-atom}$. This MAE value is comparable to the magnetic anisotropy energies ($\sim 60 \mu\text{eV}/\text{atom}$) of 2D Cr_2NO_2 and Mn_2NO_2 , which exhibit intrinsic long-range magnetic order with out-of-plane easy axes [ACS Nano **12**, 6319–6325 (2018)]. To fully address this comment, we have included the following content in the revised manuscript and added Supplementary Table 24 to the Supplementary Information:

“Additionally, we calculate the magnetic anisotropy of 2D Fe_2O_3 (101), and the results show that 2D Fe_2O_3 (101) exhibits an out-of-plane easy axis and a magnetic anisotropy energy (MAE) of 50 $\mu\text{eV}/\text{Fe-atom}$ (see supplementary Table 24).” (P. 12)

Supplementary Table 24. Total energy per Fe atom for 2D Fe_2O_3 (101) with the magnetic axis oriented in the out-of-plane (m_\perp) and in-plane (m_\parallel) directions.

Material	$E(m_\perp)$ meV/Fe-atom	$E(m_\parallel)$ meV/Fe-atom
Fe_2O_3 (101)	-17196.299	-17196.252

Comment 1.3: 3) *The authors should estimate transition temperature using more recent approaches compared to (or in addition to) the Ref. 38 they use in the paper. One way to estimate can be found in Eq. (3) of <https://journals.aps.org/prresearch/abstract/10.1103/PhysRevResearch.3.043024>*

Reply: We thank the referee for bringing to our attention the fascinating research conducted by Tiwari et al. [Physical Review Research **3**, 043024 (2021)]. This work proposes a closed-form formula to estimate the Curie temperature (T_C) of 2D ferromagnetic (FM) materials for all common 2D lattice types. We comply with the referee’s suggestion and adopt an analogous approach to predict the Néel temperature (T_N) for 2D antiferromagnetic (AFM) Fe_2O_3 . Specifically, according to the mean field theory, $3k_B T_N = S(S + 1) \sum J$ [Solid State Communications **219**, 25-27 (2015); Physical Review B **72**, 214105 (2005)], where k_B is Boltzmann constant, S is the spin eigenvalue, and J is the nearest-neighbor exchange strength. The exchange strength J is obtained from non-collinear DFT total energy calculations using the Eq. 4(a-c) in Ref. [Physical Review Research **3**, 043024 (2021)]. By utilizing the above parameters, we can estimate that the Néel temperature of our predicted 2D AFM Fe_2O_3 is equal to 700.61 K, which is comparable to our previous estimated T_N value of 506.5 K using the method in previous Ref. [38] that is replaced by the current estimation.

To fully address this comment, we have updated the contents regarding the estimation of Néel temperature for Fe_2O_3 in the revised manuscript, as shown below:

“Further, we employ the mean field theory to estimate the Néel temperature of 2D Fe_2O_3 (101), using the formula $3k_B T_N = S(S + 1) \sum J^{38,39}$, where k_B represents the Boltzmann constant, S is equal to 5/2 due to d^5 for Fe^{3+} , and J (3.45 meV)

denotes the exchange strength of one Fe with its six nearest neighbors. The J value is obtained from non-collinear DFT total energy calculations based on the Heisenberg model, using $J = \frac{J_{\perp} + J_{\parallel}}{2}$, where $J_{\perp/\parallel} = \frac{E_{\text{FM}}^{\perp/\parallel} - E_{\text{AFM}}^{\perp/\parallel}}{2N_{\text{NN}}S^2}$ ⁴⁰. Here, $E_{\text{FM}}^{\perp/\parallel}$ and $E_{\text{AFM}}^{\perp/\parallel}$ represent the total energies for FM and AFM order with the magnetic axis oriented in the out-of-plane/in-plane direction. N_{NN} refers to the number of nearest neighbors. To obtain parameters beyond the nearest neighbor, one can consult the method described in reference⁴¹. Based on our calculations, we can estimate that the Néel temperature of our predicted 2D Fe₂O₃ is equal to 700.61 K, indicating a stable antiferromagnetic state at or above room temperature.” (P. 12)

Reviewer #2 (Remarks to the Author):

Comment: *The authors have satisfactorily replied to all my questions, as well as performed several new calculations to address my questions. Therefore, I think that the paper can now be accepted for publication.*

Reply: We are delighted to have *satisfactorily* addressed all of the second referee's questions, and sincerely thank the referee for recommending our work for publication in Nature Communications.

Reviewer #3 (Remarks to the Author):

Comment: *The authors have provided a set of comprehensive studies and analyses based on the referee's comments. All the concerns have been addressed and the conclusions are supported by additional data. I would therefore recommend the publication of this interesting material discovery work in Nature Communications.*

Reply: We express our heartfelt gratitude to the third referee for his/her highly positive assessments and for the recommendation of our work to publish on Nature Communications.